# Ethnomedicinal and Ethnobotanical Survey in the Aosta Valley Side of the Gran Paradiso National Park (Western Alps, Italy)

**DOI:** 10.3390/plants11020170

**Published:** 2022-01-09

**Authors:** Cristina Danna, Laura Poggio, Antonella Smeriglio, Mauro Mariotti, Laura Cornara

**Affiliations:** 1Department of Earth, Environment and Life Sciences, University of Genoa, Corso Europa 26, 16132 Genoa, Italy; m.mariotti@unige.it; 2Forest Botanical Conservation Office, PNGP, Paradisia Alpine Botanic Garden, Valnontey 44, 11012 Aosta, Italy; laura.poggio@pngp.it; 3Department of Chemical, Biological, Pharmaceutical and Environmental Sciences, University of Messina, Viale F. Stagno d’Acontres 31, 98166 Messina, Italy; antonella.smeriglio@unime.it

**Keywords:** Cogne valley, Valsavarenche valley, Rhêmes valley, BioRefugia, cultural heritage, traditional knowledge, medicinal plants, human well-being

## Abstract

Most of traditional knowledge about plants and their uses is fast disappearing because of socio-economic and land use changes. This trend is also occurring in bio-cultural refugia, such as mountain areas. New data on Traditional Ethnobotanical Knowledge (TEK) of Italian alpine regions were collected relating to three valleys (Cogne, Valsavarenche, Rhêmes) of the Gran Paradiso National Park. Extensive dialogues and semi-structured interviews with 68 native informants (30 men, 38 women; mean age 70) were carried out between 2017 and 2019. A total of 3918 reports were collected, concerning 217 taxa (including 10 mushrooms, 1 lichen) mainly used for medicinal (42%) and food (33%) purposes. Minor uses were related to liquor making (7%), domestic (7%), veterinary (5%), forage (4%), cosmetic (1%) and other (2%). Medicinal plants were used to treat 14 ailment categories, of which the most important were respiratory (22%), digestive (19%), skin (13%), musculoskeletal (10%) and genitourinary (10%) diseases. Data were also evaluated by quantitative ethnobotanical indexes. The results show a rich and alive traditional knowledge concerning plants uses in the Gran Paradiso National Park. Plants resources may provide new opportunities from the scientific point of view, for the valorization of local products for health community and for sustainable land management.

## 1. Introduction

The traditional uses of plants express the resilient relationship between human communities and their environment. This cultural heritage, cumulated and evolved living in close contact with nature is fast disappearing owing to socio-economic and land use changes [1]. In areas historically exposed to very few external influences and centered around a subsistence economy, such as islands and alpine regions, this trend is less noticeable. Therefore, these areas represent important BioRefugia for the conservation of the biodiversity of plants and animals as well as for the cultural differences [2,3,4].

In recent years, several ethnobotanical investigations have been focused on different North Italian alpine regions, such as Piedmont [5,6,7], Lombardy, including the Stelvio National Park, [8,9,10,11,12] and Western Liguria [13]. The ethnobotanical traditions of Valle Orco, located in the Piedmont side of Gran Paradiso National Park, was previously investigated [14]. Regarding the Aosta Valley, previous ethnobotanical studies are limited and dated (e.g., the scientific studies done by Binel (1972) [15] on the traditional use of some local species and by Chimenti Signorini and Fumagalli (1983) [16] on the medicinal plants used in Valtournanche). However, several informative books concerning the same topic are known [17,18,19,20,21].

The Aosta valley, located at the north-western end of Italy, is the smallest region the country, with an extension of 3620 km^2^. The region is surrounded by mountains (Gran Paradiso, Cervino, Monte Rosa and Monte Bianco) that separate it from Piedmont, Switzerland, and France. The Aosta Valley was inhabited since the Neolithic and, starting from 25 BC, it was annexed to the Roman Empire, due to its strategic position. After the fall of the Roman Empire, the region suffered various invasions and dominations and, in 575 AD, it fell under the Franco-Burgundian kingdom, that marked the passage from a Celtic-Ligurian-Latin culture to a Franco-Roman one [22]. After dissolution of the Carolingian empire and various dominations, from the XI century the region was annexed to the Savoy dominio. The French was adopted in the Aosta Valley from the 16th century and elected as official language. Even after the Italian unification (1861), the Aosta Valley tried to preserve its peculiar linguistic and cultural traditions, which made it an autonomous bilingual region with a special statute (from 1948). The original population of the Aosta Valley commonly uses the *patois* dialect, which is clearly influenced by the Franco-provençal language [23]. The originality of *patois* lies in its variety, since there are many dialect inflections according to the valleys, neighboring municipalities, and villages.

Aosta Valley, due to its partial geographical and socio-cultural isolation, represents an ideal breeding ground for ethnobotanical research. Typical traditions, representative of the original population, are still preserved in these areas, being scarcely affected by external influences. We collected data concerning the TEK of the three valleys (Cogne, Valsavarenche, Rhêmes) included in the Gran Paradiso National Park, an area of high naturalistic interest. Our survey combined ethnobotanical data concerning the relationships between people and plants with folk medicine, to preserve the traditional uses of the local flora and to revitalize the strong cultural identity of the alpine valleys. The recovery of TEK has an intrinsic cultural value and mainly provides some new opportunities for a sustainable land management and for the valorization of local products [24]. In addition, data obtained on traditional use of natural products could be useful for the rational development of new medicines for health community.

## 2. Results

A total of 68 informants (30 men and 38 women) aged between 36 and 92 years (mean age 70 years) were interviewed in Cogne, Valsavarenche and Rhêmes valleys (Table 1), respectively.

Almost all informants (97%) have always been residents of their respective valleys, about 54% (37) of them had a primary level of education, 19% (13) have reached middle school, 22% (15) reached secondary school and 4% (three) obtained university studies. Regarding their employment, 66.2% of them were retired, previously housewives or employed as farmers, breeders, cheesemaker, veterinary, teachers, park guards, nature guides, restaurateurs, public servants, etc.

### 2.1. Plants Diversity Analysis

At the end of the interviews, we collected information on 220 plants (69 families), 10 mushrooms (six families), one lichen. In addition to the wild species (149), cultivated (41) and purchased (29) plants were also considered. As reported in previously ethnobotanical studies concerning the Italian Alps [6,8,9,13], the most quoted families (>10 species) were Asteraceae, with 26 species, followed by Rosaceae (15) and Lamiaceae (11). We also found an important role of Poaceae and Apiaceae, with 10 species, followed by Fabaceae (9). For all cited taxa, detailed data are provided in Table 2.

The final database included 3918 reports concerning 231 taxa. The most representative uses were medicinal (42%) and food (33%) followed by other categories, as reported in Figure 1.

The general importance of the useful plants in the investigated area was evaluated calculating EI and EPI indices. According to Vitalini et al. (2013) [9], the Ethnobotanicity Index (EI) was calculated as the ratio between the number of the wild taxa cited in the medicinal, cosmetic, veterinary, and food (alimentary and liquoristic) sectors and the estimated number of taxa in the wild flora of the area. The resulting value of EI (12.8%) falls above the range of values (5.37–10.75%) reported by Guarrera et al. (2008) [25] for different Italian regions and above those referred in different Italian Alpine areas (e.g., 6.2% for Val San Giacomo [9], 9.7% for Stelvio National Park [10], 11% for North-Western Ligurian Alps [13] and 12% for South Tyrol [26]).

The richness of popular knowledge about the wild species was verified by the Ethnophytonomic Index (EPI), higher (EPI 0.11) than the EPI value previously reported in Italy and for another alpine areas (EPI 0.06) [9]. Nevertheless, this value shows an erosion of the linguistic heritage associated with plants, suggesting that only 11% of the wild taxa have a vernacular name.

The local importance of each species was calculated by using the Relative Frequency of Citation (RFC). In Table 3 the species that obtained an RFC > 0.50 were reported.

*Peucedanum ostruthium* showed the highest RFC value (0.97), being reported by almost all informants (233 citations). Imperatoria, called *agrù*, is considered a panacea to cure all diseases and is mainly used to treat different medicinal and veterinary problems. Thanks to its antiseptic and anti-inflammatory properties, it is used to treat external and internal diseases. Different parts of the plant (flowers, leaves, roots) and different herbal preparations (infusion, decoction, compress) are used. Imperatoria is also used for liquoristic and for domestic purposes; roots fumigations are also reported to disinfect the stables, a practice mainly important during calving.

*Urtica dioica* (0.91), *Blitum bonus-henricus* (0.90) and *Bistorta officinalis* (0.87) are reported by most informants, especially for alimentary uses. Stinging nettle is included in seven categories of use, being reported as food and medicine for humans and animals, as cosmetic and for domestic and other purposes.

Another important plant used is *Juniperus communis* (RFC 0.88), that is reported as seasoning, for its medicinal and veterinary applications and for domestic and handcraft uses.

At the opposite side, species cited by only few informants include plants, whose traditional use has been almost completely lost (e.g., *Teucrium chamaedrys* (RFC 0.10), *Alchemilla xanthochlora* (RFC 0.09) *Geranium robertianum* and *Hylotelephium maximum* (RFC 0.07), *Euphrasia officinalis* subsp. *rostkoviana* (RFC 0.06), *Veronica fruticans* (RFC 0.03)). Even some poisonous plants, today completely disused, were reported by few informants as vermifuge or as abortifacient e.g., *Hedera helix*, *Daphne mezereum*, *Dryopteris filix-mas* and *Juniperus sabina* (RFC 0.02).

### 2.2. Medicinal Uses

Medicinal plants represent the most important category of use, with a total of 1639 citations concerning 124 taxa, including 122 plants belonging to 51 families, one lichen (Parmeliaceae) and one fungus (Agaricaceae). Asteraceae (20 species), Lamiaceae (eight), Rosaceae (eight), Pinaceae (five), Plantaginaceae (five), Gentianaceae (four) and Apiaceae (four) are the families with the highest number of species.

Most used parts were flowers and leaves (30%), followed by fruits and seeds (17%), roots (11%), aerial parts (7%), resin, latex and sap (4%). Major herbal preparations included infusion (35%), decoction (30%), syrup (11%), poultice/compress (10%) and maceration in oil (7%) or alcohol (7%).

Human disorders were classified into 14 categories based on the International Statistical Classification of Diseases and Related Health Problems (ICD-10) by the World Health Organization [27]. Used subcategories concern diseases of the: respiratory tract; digestive system; subcutaneous tissues; genitourinary tract; musculoskeletal system and connective tissue; circulatory system; nervous system; eye and adnexa; ear and mastoid process (sensory system); infections and parasitosis; pregnancy, childbirth and puerperium; symptoms and signs not elsewhere classified; dental and oral; endocrine, nutritional and metabolic.

As reported for other alpine areas [9,13], also in the PNGP, the most frequent treatments were those of the respiratory, digestive, and integumentary systems, showing the highest citations (352, 315 and 219, respectively), followed by those of the genitourinary tract, musculoskeletal system and connective tissue (169 and 156 citations, respectively) (Table 4).

The informant consensus factor (Fic), calculated for each medicinal subcategory, ranged from 0.53 to 0.90. The obtained values showed a good consensus on the choice of plants for most disease categories, with values close to 1 (from 0.80 to 0.90) (see Table 3). For the treatment of dental and circulatory problems, for the treatment of problems related to pregnancy, for infectious and parasitic diseases and nervous problems a greater choice of medicinal plants has been reported (Fic from 0.69 to 0.75).

Considering the use of plants in relation to specific disease categories, we calculated the Fidelity Level (FL) and selected the species with a FL > 70% and with at least 10 citations in the disease category for which the highest FL has been obtained. In Table 5 we reported these species with the related disease category and their main bioactive compounds involved in the specific therapeutic effects based on pharmacological investigations. 

Some of the most important medicinal plants are depicted in Figure 2A–G. *Peucedanum ostruthium* (Figure 2A) is not included among species with high FL, because it is used to treat several categories of diseases, including respiratory, digestive, muscular, skin and genitourinary problems.

For the respiratory tract, the species with the highest FL value (100%) were *Cetraria islandica* together with *Tussilago farfara* (91.7%) (Figure 2B) and *Viola calcarata* (98.3%) (Figure 2C), used to treat phlegm, bronchitis and colds. Other species used for the same purpose are *Artemisia umbelliformis*, *Gymnadenia nigra*, *Teucrium chamaedrys*, *Thymus pulegioides*, *Salvia officinalis* and *Artemisia genipi*. In addition, different Pinaceae such as *Pinus sylvestris* L. (Figure 2D), *Pinus cembra* (Figure 2E), *Pinus mugo* and *Picea abies* were widely used for the treatment of respiratory diseases: young resinous cones or buds were macerated with sugar in a glass jar and exposed to the sun to obtain a syrup. For the digestive system high FL value were found for *Carum carvi*, *Gentiana punctata* and *Juniperus communis* (Figure 2F). For the diseases of the skin and subcutaneous tissues, leaves of *Calendula officinalis*, *Plantago* spp., *Hilotelephium maximum* were widely used, while latex of *Chelidonium majus* and *Euphorbia seguieriana* were indicated to treat warts. For the diseases of the genitourinary tract, *Arctostaphylos uva-ursi* was the most quoted plant followed by *Vaccinium vitis-idaea* and *Polygonum aviculare*. Among the species referred for the treatment of musculoskeletal system, *Arnica montana* (Figure 2G) was used to treat blows and external hematomas, while *Equisetum arvense* in form of decoction to strengthen bones and against osteoporosis. For the diseases of the sensory system (eye and adnexa), the most important *taxa* were *Rosa* sp., *Euphrasia officinalis* subsp *rostkoviana* and *Matricaria discoidaea*. Infections and parasitosis were commonly treated with flowers of *Tanacetum vulgare* and bulbs of *Allium sativum*. For pregnancy, childbirth and the puerperium *Linum usitatissimum*, *Malva neglecta* were used as emollients, while *Carduus defloratus* L. was reported as a galactogogue.

### 2.3. Liquoristic Uses

Different liqueurs consisting of herbal macerates in alcohol were considered traditional preparations with also medicinal properties. In the studied area, 35 different taxa (35 plants and 1 fungus) were reported in liqueur making.

Digestive liqueurs and grappas were prepared with artisanal systems (Figure 3A,B). The roots of *Gentiana punctata* (Figure 3C) and *G. lutea*, as well as the flowers of *G. acaulis* and *G. verna* (Figure 3D) were reported for this purpose. The aerial parts of different species such as *Achillea erba-rotta* (Figure 3E) and *A. moschata* were added in the *Fernet* liqueur; *Artemisia genipi* (Figure 3F), *A. glacialis*, *A. umbelliformis* in the *Genepì* liqueur. The *Kummel* liqueur was made by using the seeds of *C. carvi*, while the *Arquébùse* using the leaves of *T. vulgare*. In addition, several species belonging to the Pinaceae were quoted to flavor grappas and liqueurs (e.g., *Larix decidua*, *Picea abies*, *Pinus cembra*, *P. mugo* and *P. sylvestris*). Even wild berries were used to flavor grappas (e.g., *Rubus idaeus*, *Vaccinium myrtillus* and *Rosa canina*). In the Cogne valley, a traditional beer recipe obtained by macerating in water the seeds of *Hordeum vulgare* L., the fruits of *Berberis vulgaris*, and the roots of *Polypodium vulgare*, was referred. Lastly, several informants reported the old use of the fungus *Fomitopsis officinalis* (Figure 3G), added in liqueur for its bitter properties.

### 2.4. Food Plants and Edible Fungi

Alimentary plants represent the second most important category of uses, with a total of 1302 citations concerning 103 edible species belonging to 40 families. Rosaceae (10 species), Asteraceae (eight), Lamiaceae (six), Fabaceae (six), Poaceae (six), Ericaceae and Grossulariaceae (five) are the most represented families. Data about nine edible fungi belonging to five families were also collected; Agaricaceae, Boletaceae and Suillaceae represent the most quoted families.

The leaves of *Bistorta officinalis*, *Blitum bonus-henricus*, *Urtica dioica* (Figure 4A,B) and *Taraxacum officinale* were used as ingredient in soups, in omelets or stir-fried with butter and eaten as a side dish. The young leaves of *Taraxacum officinale* were also often consumed raw in salad, with boiled eggs, potatoes and walnut oil. Other wild species less frequently reported as ingredients for soups were *Silene vulgaris*, *Tragopogon pratensis*, *Primula veris*, *Phyteuma* spp. and *Rumex acetosa*. *Botrychium lunaria* (Figure 4C) is a very valued fern used in summer’s soups, especially in the mountain pastures.

Several species were mainly eaten in the past as a snack. For example, the leaves of *Rumex acetosella* and the stems of *T. pratensis* for their refreshing properties; the root of *P. vulgare* for its sweet taste of licorice, while bulbs of *Bunium bulbocastanum* (Figure 4D) for their chestnut flavor. Edible flowers cited by informants included *P. veris*, *Viola calcarata* and *Trifolium pratense*.

Various wild or cultivated fruits were eaten fresh or used for jelly, jams or syrups. The most quoted included *Fragaria vesca*, *Ribes nigrum* and *R. rubrum*, *Rubus idaeus* and *Vaccinium myrtillus*, followed by *Amelanchier ovalis* (Figure 4E), *Berberis vulgaris*, *Hippophae rhamnoides*, *R. canina* (Figure 4F), *Arctous alpina*, *Ribes uva-crispa* and *R. petraeum*, *Rubus saxatilis*, *Sambucus nigra*, *Vaccinium uliginosum* and *V. vitis-idaea*.

Among the aromatic plants, some are purchased form valley floor such as *Laurus nobilis*, *Salvia rosmarinus* and *S. officinalis*, while others are wild local species such as *T. pulegioides* and *Juniperus communis*, used for seasoning meat (e.g., in the *mocetta* and in the ’*suede in civet*’ recipes). In addition, some exotic species were also used as spices (e.g., *Myristica fragrans* and *Syzygium aromaticum*). All these aromatic plants, with the addition of *C. carvi* fruits, were used in the past for seasoning the *repùta*, a typical recipe for storing some cultivated vegetables (*Allium ampeloprasum, B**rassica rapa*, *Brassica oleracea*, *Daucus carota* and *Beta vulgaris*). *Hordeum vulgare*, *Secale cereale* and *Triticum* sp.pl. were widely cultivated in the past to make flour for baking. *Solanum tuberosum* is also today an important cultivated plant, playing an important role in human nutrition in the mountain areas. The most common vegetable rennet used in the past in cheesemaking were *Ranunculus* sp.pl., *Galium* sp.pl., *Lotus corniculatus, Urtica dioica, Bistorta officinalis* and *A. genipi*. Roasted roots of *Taraxacum officinale* and *Cychorium intybus* and roasted seeds of *Hordeum vulgare* were used as coffee substitutes until the second postwar period.

### 2.5. Veterinary Uses and Forage

Species of veterinary interest account for 5% of the total uses, with 229 citations, while the plants used as fodder account for 4% with 123 citations. Altogether, data for about 61 species belonging to 29 families were collected. The most represented families were Asteraceae (eight species), Fabaceae (seven) and Poaceae (seven), followed by Apiaceae, Rosaceae and Polygonaceae (four species).

The daily cows feeding was made up of hay, consisting of *Festuca* sp. added with some cultivated plants such as *Onobrychis viciifolia* and *Medicago sativa*. In addition, the mash given to fatten animals and increase milk production included *Brassica rapa*, *Solanum tuberosum*, *Triticum* sp., *Zea mays* and *Avena sativa*; *Hordeum vulgare* is also added for its anti-inflammatory properties.

Several wild plants, common at lower altitude pastures, were referred as good fodder for cows (e.g., *Bistorta officinalis*, *Alchemilla vulgaris*, *Taraxacum officinale*, *Trifolium pratense* and *T. repens*). In the higher altitude pastures, other plants increased the milk production and its quality (e.g., *Trifolium alpinum* and *Viola calcarata*).

Several remedies are reported in case of cow pregnancy. First, 15 days before the calve birth, some emollient plants were given in form of decoction such as *Linum usitatissimum* and *Malva neglecta* or *M. sylvestris* L. Even the pods of *Vicia faba* and *Phaseolus vulgaris* were reported for their emollient properties. The decoction prepared with *Achillea millefolium* was given to the pregnant cow as an anti-inflammatory, while the flowers of *Carduus defloratus* and *Carlina acaulis* were added to the mash to increase the milk production.

Several plants were used to treat different animal diseases (e.g., for the digestive system problems, the roots of *Gentiana punctata* or *G. lutea* or the berries of *J. communis* were added in the fodder mixture). *Arnica montana* and *P. ostruthium* were reported to treat musculoskeletal problems; Imperatoria was also used to treat skin problems such as infected wounds and hoofing problems. *Mentha* sp. was reported in case of cow mastitis.

The leaves of *Taraxacum officinale*, *Trifolium pratense* and *Urtica dioica* were also reported as a good fodder for rabbits and hens. During the winter nettle was added dried in the fodder to increase eggs production.

As repellents for noxious insects, leaves of *Levisticum officinale* and *Veratrum album* were indicated.

### 2.6. Domestic and Handcraft

In this study, 340 citations were related to plants used for domestic and handcraft purposes, including 39 species belonging to 28 families. Among these, Apiaceae, Lamiaceae, Pinaceae and Salicaceae are the most represented with three species.

Several species were used for the manufacture of small tools and as construction wood. *Larix decidua* and *Picea abies* were used for construction purpose such as to build roofs and perimeter walls, respectively. On the contrary, thanks to its durable wood, *Fraxinus excelsior* L. was used to make blades and pickaxes (Figure 5A). *Pinus cembra* was considered the most valuable wood for building furniture, because of his resinous smell and his mothproof power. *Juniperus communis* was used for building the stick to turn the polenta, to which confers its typical aroma. Brooms made by branches of *Betula pendula* Roth and *B. vulgaris* were used to clean the stables and to remove the wheat chaff from the barn floor. The branches of *Sambucus nigra*, *Viburnum lantana*, *Corylus avellane*, *Clematis vitalba*, *Salix caprea* and *S. purpurea* were used for making baskets such as the traditional *gerle* (Figure 5B). In the past, the cortex of *Betula pendula* was used to build handcrafted snuffboxes with artistic inlaid designs (Figure 5C). Domestic uses of plants for *Valeriana celtica* and *Lavandula angustifolia* put in the closets against moths were also referred.

### 2.7. Others Uses

During the interviews, several other uses and traditional practices related to the local flora were collected. The aqueous macerate of *Urtica dioica* was widely used as fertilizer and as insecticide in home gardens. Moreover, *Carlina acaulis* and *Stipa pennata*. (Figure 5D) were used as timepiece plants, hanged on the houses for weather forecast. Flowers of *Leontopodium alpinum* Cass. (Figure 5E) were collected, dried and used as bookmarkers. Some plants were used in the past for ludic activities (e.g., the flowers of *Arctium lappa* were thrown on clothes by children; whistles were made by using the hollow stems of *Heracleum spondhylium* and *Angelica silvestris*). Several plants were also reported as tobacco substitutes (e.g., the leaves of *Arnica montana* and the needles of *Pinus* spp).

Several plants were traded until the second postwar period as a source of income: the aerial parts of *Artemisia genipi*, *A. absinthium*, *Achillea erba-rotta* and *A. millefolium;* the roots of *Gentiana lutea* and *G. punctata*; the thallus of *Fomitopsis officinalis*.

During the interviews, some traditional magico-spiritual practices, were documented: in the Epinèl village, in the Cogne Valley, during St John fest (24 June), a bunch of wildflowers was collected at dawn, still covered in dew, and placed on the house doors in sign of protection. Similar practice was found in the Rhêmes valley, where the flowers of *Paradisea liliastrum* (Figure 5F) were used to build a cross with flowers protruding at the ends that was placed on the doors against the misfortune.

Other uses not related to plants were recorded in the studied area. In particular, the marmot fat and the ibex marrox were indicated to treat respiratory and musculoskeletal diseases. In these valleys, the presence of local healers called *Rabeilleurs* showing therapeutic ability in massage in case of musculoskeletal problems, were also documented. In addition, in Aosta valley still persists the practice of *Sécrèts*, consisting of the treatment of several diseases (e.g., warts, worms, burns, pains and others) by using prayers and rituals. The healing formulas were kept secret and handed down mainly orally, usually within the family or the village [48].

## 3. Discussion

The present results are consistent with ethnobotanical data from other Italian alpine areas [5,6,7,8,9,10,11,12,13,15,16]. The high EI value obtained in our study indicates that the knowledge of useful plants is still well consolidated in the studied area. The impervious territory and the climate, characterized by severe and long winters, forced the population to collect abundant supplies of food, medicinal herbs and firewood suitable for survival. As food reserve, in addition to cultivated cereals and potatoes, other vegetables such as cabbages, carrots, turnips and leeks were cultivated in home gardens. Elderly people remember a typical dish, called *repùta*, that was prepared with these fermented vegetables in the Cogne valley. Nowadays, in home gardens, in addition to food plants, several aromatic and medicinal plants were also commonly cultivated (e.g., *Calendula officinalis*, *Levisticum officinale*, *Cyanus segetum*, *Tanacetum vulgare* and *Rosa × alba*). These species were dried and stored or used fresh to prepare traditional remedies such as infusions, liqueurs and syrups.

An important source of food and medicine also came from wild plants and fungi. According to data from other Italian alpine regions, many small edible fruits and berries such as different *Vaccinium* and *Ribes* species are still today collected [5,6,7,8,9,10,11,12,13,15,16]. However, in the studied area, our informants also cited the use of other fruits: *Arctous alpina* and *Rubus saxatilis*. As regard to edible wild plants, used raw in salads or boiled, besides the most common species collected in the alpine areas, data concerning the Pteridophyte *B. lunaria*, used to prepare tasty soups, were also recorded. Curiously, this fern was also reported in the neighboring Waldesian valleys of Piedmont for the treatment of skin diseases [7].

Many wild plants were also quoted for their medicinal properties, used alone or in association to treat several diseases. Traditional remedies were used for the treatment of respiratory diseases, very frequents among the local population. Among species quoted for the treatment of cough, cold and flu, in addition to some *Viola* species, reported also in other alpine zones, the use of *V. calcarata*, was recorded. Lamaison et al. (1991) [36] found in this species different bioactive compounds such as flavonoids, rutin and mucilages, that could be related to its activity against airway problems. Indeed, the effectiveness of quercetin-type flavonols such as rutin against viral low respiratory tract infections has been recently demonstrated [49]. The traditional use of the lichen *C. islandica* for the treatment of respiratory diseases is probably due to the presence of polysaccharides with antiviral [28] and anti-inflammatory properties [29]. For the same purposes, also several *Artemisia* species are used. In this regard, Vouillamoz et al. (2015) [50], suggested that *Artemisia* bioactive compounds (sesquiterpene lactones) could be involved in the activation of bitter receptors, stimulating ciliary motion and relaxation of bronchial tissues. By this way, these compounds prevent infections and improve ventilation; the absorption path of these volatile compounds explains the use of inhalations in addition to herbal tea, the main preparation traditionally used.

The use of different *Pinaceae* species is widespread throughout the Alps to prepare antitussive syrups with expectorant and decongestant properties [7,8,9,10,11,12,13,16]. Phenolic acids, flavonoids, proanthocyanidins, terpenoids and resin acids with antimicrobial and anti-inflammatory activities can contribute to their healthy effect on the respiratory system [30,34,35,37].

Digestive diseases are also very common in the studied area, probably related to a diet rich in animal fats. In agreement with other investigation on the traditional medicine of Alpine areas [5,6,7,8,9,10,11,12,13,15,16], the digestive use of different species of *Gentiana* (mainly *G. punctata*), *Achillea* and *Artemisia*, as well as of *J. communis* and *C. carvi*, were recorded. *Artemisia* spp., representing iconic plants of the Alpine region, were commonly called *génépi* and they were widely used for infusions and liqueurs making with thermogenic and digestive effects [50].

Skin and musculoskeletal problems often affected the population involved in manual and field work and consequently several plants were cited for this purpose. The use of *Arnica montana* for musculoskeletal diseases was related to the presence of helenalin and dihidrohelenalin-type sesquiterpene lactones, flavonoids and phenolic acids with anti-inflammatory activity [43,44,45]. On the contrary, *Plantago* spp. were widely used being rich in polysaccharides and flavonoids with anti-inflammatory effects for treatment of wounds and other skin problems [40]. Some species were also indicated to treat infections and parasitosis such as *Allium sativum* and *Tanacetum vulgare. A. sativum* with sulfur-containing phytoconstituents and flavonoids showed antibacterial, antiviral, antifungal, antiprotozoal, antioxidant and anti-inflammatory properties [39], whereas *T. vulgare*, rich in β-thujone, was responsible for the anthelmintic activity [33].

Regarding the medicinal and veterinary properties, *P. ostruthium* deserves a special mention, representing the core of the cultural ethnobotanical heritage of the investigated area, as showed by RFC value. The rhizome has a long tradition of medicinal use and the plant has been known as ’*Divinum remedium*’ since the eighteenth century [51]. The phytochemical profile of the rhizomes of *P. ostruthium* showed the main bioactive compounds such as coumarins, oxypeucedanin hydrate, oxypeucedanin, ostruthol, imperatorin, osthole, isoimperatorin and ostruthin [52]. In Aosta valley, the plant is considered a panacea for all ailments. Moreover, in our survey, information concerned traditional uses not only of root/rhizome but also of leaves and the flowers, used both as external and internal remedies. Similar uses of these aerial portions are only rarely referred for other alpine areas such as Swiss Alps [53], South Tyrol [26] and Austria [54].

Given the considerable economic importance of pastoralism in these valleys, several plants were quoted to increase milk production and quality. Data on some species typical of alpine pastures such as *T. alpinum* and *V. calcarata*, rarely reported in previous studies, were recorded [5,6,7,8,9,10,11,12,13,15,16]. Our informants also reported the past use of different species of *Galium* as rennet, in agreement with similar data reported by Biella and Pieroni (2015) for Piedmont. In addition, other species were cited for the same purpose in the studied area such as *L. corniculatus*, *U. dioica*, *Ranunculus* spp. and *Bistorta officinalis*.

Quantitative analysis of the data collected showed the most cited species (RFC values), representing the heart of the ethnobotanical culture of the investigated area. In addition, Fic and FL gives us useful indications about medicinal species worthy of being further studied from the phytochemical and pharmacological point of view.

The general importance of the useful plants in the study area was showed by a very high EI value, compared with those obtained by studies on other Italian alpine areas and such as another protected area, Stelvio National Park [10]. Similarly, EPI value obtained in our survey was higher than that referred for Stelvio National Park, showing a good popular knowledge concerning local names of the wild species in PNGP. Although in the Aosta Valley the *patois* dialect is very widespread, several differences in the plant dialectal names among the three valleys indagated, were found. Moreover, the documented dialect names used in Aosta Valley were similar to those used in Lower and Central Valais (Switzerland) (e.g., *J. communis* called *tsénèvro* in Aosta Valley and *dzenièvro* in Switzerland, *Artemisia vulgaris* reported as *porta rusò* and *porta-rozò* and *V. vitis-idea* known as *gravelòn* and *gravèlong*, respectively) [53]. This knowledge is fast disappearing because of the linguistic homologation that characterizes modern societies. Therefore, the documentation of these local names is crucial also to preserve TEK handed down orally from one generation to the next.

## 4. Materials and Methods

### 4.1. Study Area

The investigated area included the three valleys, namely Cogne, Valsavarenche, Rhêmes, of the Gran Paradiso National Park (PNGP) in Aosta Valley, Western Alps, Northern Italy (Figure 6). In addition, the land use and vegetation map of the PNGP is added as Appendix A. [55]

The PNGP, previously a royal hunting reserve ceded to the state at the beginning of the twentieth century by the King Vittorio Emanuele III of Savoy, is a protected area established in 1922. It is the Italy’s first National Park established to preserve fauna, flora and the natural beauty of the landscape. The PNGP covers 71.044 ha between Aosta Valley (52%) and Piedmont (48%) and includes five valleys (Cogne, Valsavarenche, Rhêmes, Orco, Soana) surrounding the Gran Paradiso peak (4061 m). The current geomorphological set-up is the result of the action of the glaciers that led to the formation of large valleys, while currently, the main modeling agent of the landscape is river engravings. The cliffs are mostly of siliceous origin, although there are some areas of calcareous and limestone, especially on the Aosta Valley side. The lithology, geomorphological and climatic features strongly influence the vegetation. The great biodiversity of the park is due—in addition to the considerable extension of its territories—to the presence of two slopes, the Aosta Valley and the Piedmont ones, very different in lithology and climatic characteristics. Furthermore, the landscape is characterized by several micro-environments due to the significant differences in altitude (from 800 m to 4061 m a.s.l.). The protected area has an average altitude of about 2400 m and the subalpine, alpine and snowy vegetation types are the most represented natural environments. Forest formation mainly consists of *Larix decidua* and *Picea abies*, with *Pinus cembra* at the higher altitudes; in alpine grasslands *Festuca* sp., *Carex curvula* and *Sesleria caerulea* are the most represented species. The flora of prairies and pastures is mainly composed of acidophilous species, due to the wide dominance of the siliceous substrates; however, in calcareous outcrops also basophilous species are present.

Until today, about 1160 taxa [56], including Lycopods, Horsetails, Pteridophytes, Gymnosperms and Angiosperms, have been listed within the PNGP, 82 of which are endemic to the Alps. Native species account for 99% of the total (1127 species). Due to the high degree of plant biodiversity and to the high number of rare species, the entire territory of the park has been declared Site of Community Importance (ZSC IT 1201000). Collection of plants within the National Park is subject to the regulation-excerpt of the PNGP [57]. The gathering of plants is currently prohibited unless specific authorization. However, only for residents, a few edible and medicinal species with a strong cultural value are excluded.

In the 13 Municipalities of the Park, including 6 municipalities in Piedmont and 7 in Aosta Valley, live about 8300 people. Our research was focused on the municipalities of Cogne (1370 inhabitants), Valsavarenche (169 inhabitants) and Rhêmes (251 inhabitants), representing the Aosta Valley area belonging to the PNGP. The dialectal language commonly used by all the residents of the three valleys is called *patois*.

### 4.2. Field Study and Data Collection

Ethnobotanical data were collected during summer over three consecutive years (2017–2019), through extensive dialogues and semi-structured interviews with 68 inhabitants aged between 36–92 years (mean age 70 years). Informants were native or longtime residents in the area and had strong links with the traditional human activities of the territory. We selected informants using snowball techniques and we tried to ensure that all key informants were interviewed. We obtained the oral prior informed consent from all informants, according to the ISE (International Society of Ethnobiology) Code of Ethics. During the individually or in groups interviews, we tried to build a relationship of confidence with informants, to facilitate the dialogue. The age, gender, origin, level of education and occupation of all informants were recorded.

The questions were aimed at documenting the use of plants as food and medicine for humans and animals. In addition, also liquoristic, domestic, cosmetic and others uses were documented. The informants were asked to provide local name, parts used, period of gathering, association with other plants, preparation and use, related recipes and further indications. We reported the plant uses derived from the oral tradition in the local community. During the interviews we collect several fresh plants and dried samples representative of the local officinal flora. The nomenclature of plants follows “*Plants of the world*” [58] and the corresponding synonymous were added, according to “*Flora d’Italia*” [59,60]. “*Index fungorum*” [61] was used for the nomenclature of fungal species. Voucher specimens of the wild cited plant species were prepared and deposited at the Ethnobotanical Herbarium of the PNGP in the Paradisia Alpine Botanic Garden (Valnontey, Cogne).

### 4.3. Data Analysis and Quantitative Indices

All the ethnobotanical data were added and organized in spreadsheets of Microsoft Excel, in order to process the survey results.

Data were evaluated by quantitative parameters such as ethnobotanicity and ethnophytonomic indices, relative frequency of citation, factor informant consensus and fidelity level.

#### 4.3.1. Ethnobotanicity Index

Ethnobotanicity Index (EI) [62] allows to estimate the importance of the useful plants in a defined area. It is the ratio, expressed as percentage (%), between the number of used wild plants and the number of species making up the flora of the considered territory.

#### 4.3.2. Ethnophytonomic Index

Ethnophytonomic Index (EPI) [63] allows to estimate the wealth of the people’s knowledge about the local plant species. It is the ratio between the number of plants with vernacular name and the total number of species belonging to the studied area.

#### 4.3.3. Relative Frequency of Citation (RFC)

The local importance of each species was calculated by using the Relative Frequency of Citation (RFC) [64]:RFC = FC/N(1)
where FC is the number of informants, who mentioned the use of the species and N is the total number of informants (68 in this study).

#### 4.3.4. Factor Informant Consensus (Fic)

Factor informant consensus (Fic) [65] is used to identify the main categories of diseases and to consider the agreement among the respondents on the use of plants. It was calculated according to the following formula:Fic = (Nur − Nt)/(Nur − 1)(2)
where Nur refers to the number of use citations in each category and Nt to the number of species used in the same category. The values of Fic ranging from 0 to 1. A high value is obtained when one or few plant species are used by a large proportion of informants to treat a given disease category, while a low value indicates that the informants are in disagree about the taxa to be used in the treatment of a certain ailment.

#### 4.3.5. Fidelity Level

The Fidelity Level Index (FL) [66,67] was also considered to indicate the informants’ choice for a potential plant species to treat a given disease. It was calculated by the following formula:FL = (Np/N) × 100(3)
where Np is the number of use reports for a given species reported to be used for a particular ailment category and N is the total number of use reports cited for any given species.

## 5. Conclusions

Our survey provided an exhaustive prospect of the Ethnobotanical Traditional Knowledge (TEK) in the territory of the Gran Paradiso National Park, located in the Aosta Valley, an area so far scarcely studied from the perspective of plant folk traditions. Data collected confirmed that this knowledge mainly remains in the memories of the eldest population and enhanced the important role of the mountain areas as biocultural refugia. However, TEK is fast disappearing among the new generations and, therefore, documenting and preserving such information is crucial to reduce the loss of biocultural diversity. The results of our study contribute to the goal of Article 8(j) of the Convention on Biological Diversity (CBD) that recognizes the importance of preserving the traditional knowledge of indigenous peoples to conserve biological diversity and to ensure the sustainable use of its components. We observed that the uses of food wild plants to prepare typical dishes and liqueurs, as well as the use of medicinal plants to cure the most common diseases of human and livestock, are still well preserved by local population. Some of these species need further investigation (e.g., *B. lunaria* for its nutritional properties). In regard to medicinal plants, inhabitants give a great healing value to all the portions of *P. ostruthium* used in these valleys for the treatment of several diseases. Further studies are in progress to better characterize the phytochemical profile and the biological activity of these plants. As a future perspective, it would be useful to complete the ethnobotanical investigation also in the Piedmont side of the PNGP. It would also be useful to select some medicinal species and evaluate the possibility of local production of phytotherapeutics based on traditional remedies. The Park would play a key role in optimizing the ecosystem services of the flora, in a context of greater involvement of local populations.

## Figures and Tables

**Figure 1 plants-11-00170-f001:**
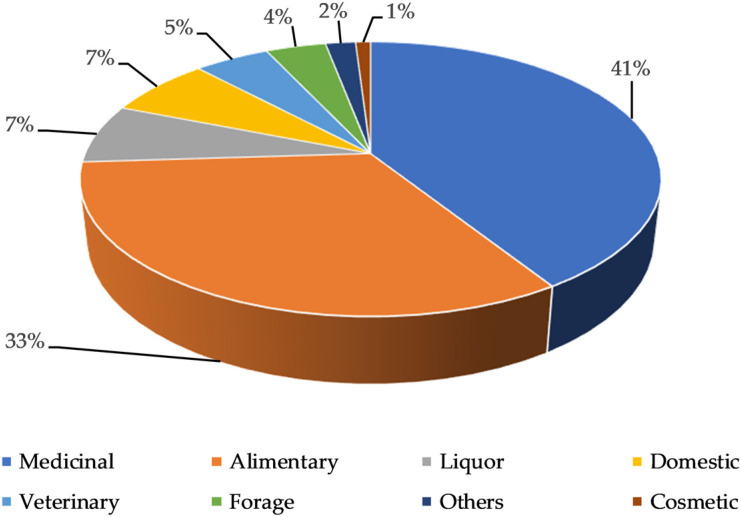
Percentage of citation and categories of use.

**Figure 2 plants-11-00170-f002:**
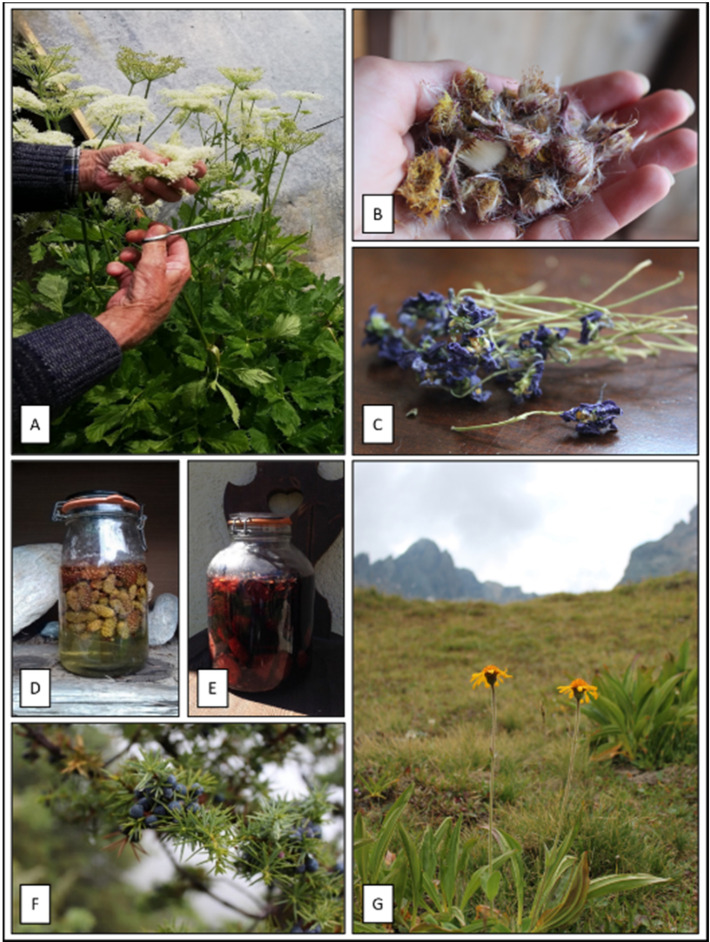
Medicinal plants: (**A**) *Peucedanum ostruthium*; (**B**) *Tussilago farfara*; (**C**) *Viola calcarata*; (**D**) *Pinus sylvestris* syrup; (**E**) *Pinus cembra* syrup; (**F**) *Juniperus communis*; (**G**) *Arnica montana*.

**Figure 3 plants-11-00170-f003:**
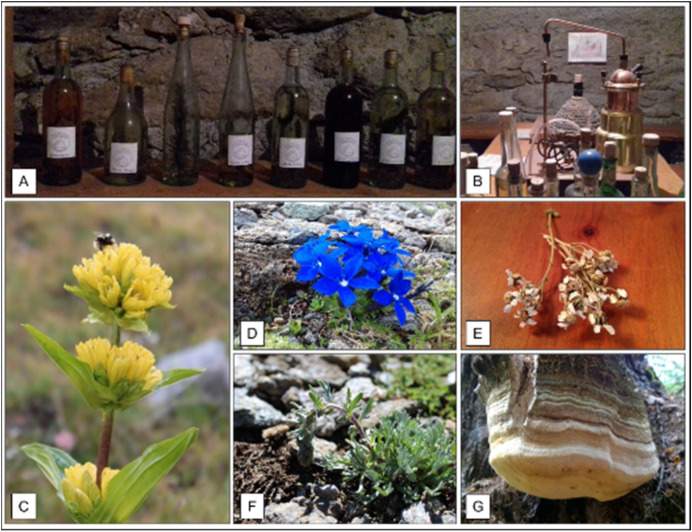
Liquoristic use: (**A**,**B**) Artisan distillery and several bottles of typical liqueurs and grappas; (**C**) *Gentiana punctata*; (**D**) *Gentiana verna*; (**E**) *Achillea erba-rotta*; (**F**) *Artemisia genipi*; (**G**) *Phomitopsis officinalis*.

**Figure 4 plants-11-00170-f004:**
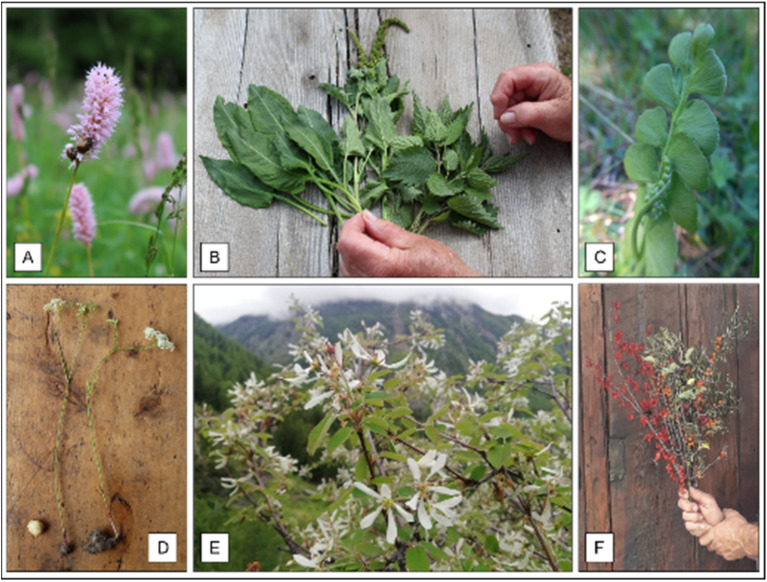
Food plants: (**A**) *Bistorta officinalis*; (**B**) *Bistorta officinalis*, *Blitum bonus-henricus*, *Urtica dioica*; (**C**) *Botrychium lunaria*; (**D**) *Bunium bulbocastanum*; (**E**) *Amelanchier ovalis*; (**F**) *Berberis vulgaris*, *Hippophae rhamnoides*, *Rosa canina*.

**Figure 5 plants-11-00170-f005:**
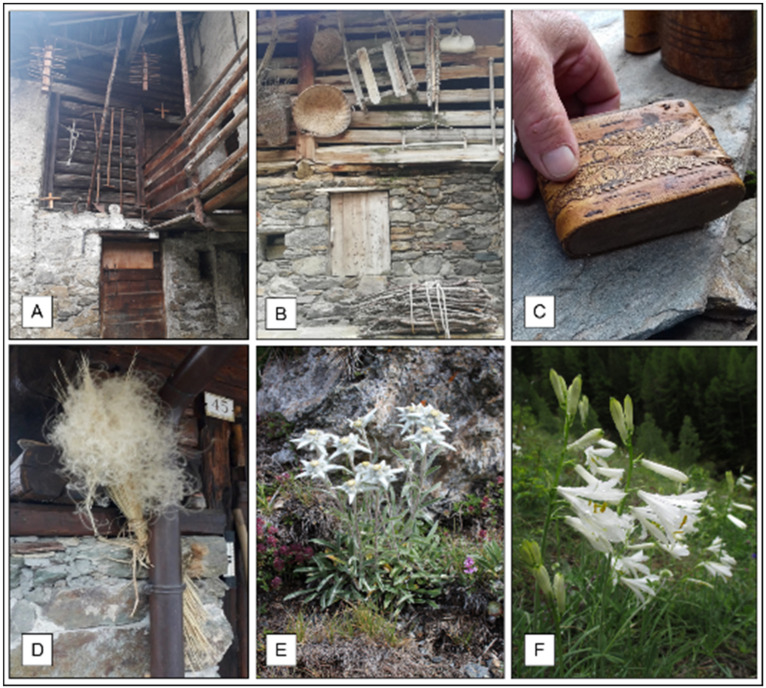
Domestic, handcraft and other uses: (**A**) Handles of tools such as shovels and picks made by *Fraxinus excelsior* wood, *ratelei*, used to store the loaves, roofs of the house made by *Larix decidua* wood; (**B**) baskets and traditional *gerle* made by *Salix* sp., other agro-pastoral tools; (**C**) snuff-singer made by *Betula pendula* cortex; (**D**) *Stipa pennata*, timepiece plant; (**E**) *Leontopodium nivale* subsp. *Alpinum*; (**F**) *Paradisea liliastrum*.

**Figure 6 plants-11-00170-f006:**
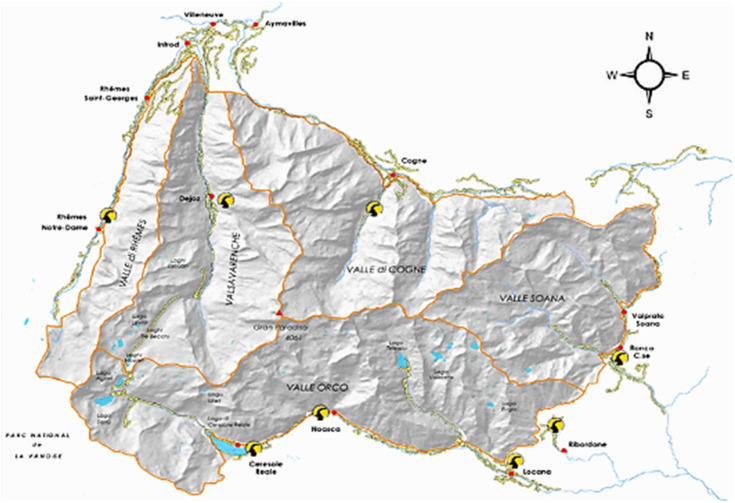
Map of the Gran Paradiso National Park.

**Table 1 plants-11-00170-t001:** Distribution of the informants, gender and residence in the municipalities of Cogne (1.544 m), Valsavarenche (1.541 m), Rhêmes (R. Saint-Georges 1.218 m and R. Notre-Dame 1.725 m).

Gender	Number of Informants and Residence Valley		
	Cogne	Valsavarenche	Rhêmes
	(1370 Residents)	(169 Residents)	(251 Residents)
men	19	4	7
women	21	9	8
total	40	13	15

**Table 2 plants-11-00170-t002:** Traditional uses of plants, fungi, lichens and others in the Gran Paradiso National Park (Aosta Valley).

Family	Vernacular and Dialectal Names ^a^	Origin ^b^	Parts Used ^c^	Ethnobotanical Uses ^d^	Similar Uses in North-Western Italian Alps ^e^	RFC ^f^
Scientific Name						
Voucher Number						
PLANTAE						
Amaranthaceae						
*Amaranthus retroflexus* L. A.ret.HBPNGP_ETN	Ertzuèino (C)	W	Leaves (fresh)	**Al**: ingredient in soups and omelettes.		0.04
*Beta vulgaris* L. subsp. *vulgaris* syn *Beta vulgaris* L. subsp. *cicla* (L.) Schübl. and G.Martens	Bietola da coste Coùte Bietola da foglie	C	Leaves (fresh)	**Al**: leaves boiled and stir-fried with butter; boiled as an ingredient in the traditional *repùta*.	[7,10,13]	0.04
*Beta vulgaris* L. subsp. *vulgaris* var. *vulgaris*	Barbabietola rossa Bètterave ròdze (C)	C	Roots (fresh)	**Al**: boiled as an ingredient in salad, russian salad and in the traditional *repùta*. **For/Vet**: used to feed cows. **Med**: recommended intake after birth (**pcp**).	**Al**: [10,13] **For/Vet**: [11]	0.20
*Blitum bonus-henricus* (L.) Rchb. syn *Chenopodium bonus-henricus* L. B.bon.HBPNGP_ETN	Spinacio selvatico Buon enrico Sepenò suvàdzu (C) Ercouvènno (C) Bon Henri (R) Pequèn (R) Verquìgno (V)	W	Leaves (fresh)	**Al**: ingredient in soups and omelettes or boiled and stir-fried with butter.	[5,6,7,9,10,11,13,16]	0.90
*Chenopodium album* L. C.alb.HBPNGP_ETN	Farinello	W	Leaves (fresh)	**Al**: ingredient in soups and omelettes.	[7,10,11]	0.03
**Amaryllidaceae**						
*Allium ampeloprasum* L.	Porro Purró (C)	C	Leaves (fresh) Roots (fresh)	**Al**: ingredient in soups and in the traditional *repùta*. **Med**: roots juice intake as vermifuge (**cip**).		0.10
*Allium cepa* L.	Cipolla Ignòn (C)	C	Bulbs (fresh)	**Al**: flavouring in several dishes. **Med**: heated onion placed on immature infections such as dental abscesses (**dent**) and painful immature boils (**skin**) (**cip**).	**Al**: [5,10,13] **Med**: [10,13] (**dent**) (**skin**)	0.03
*Allium sativum* L.	Aglio Àglie (C)	C	Bulbs (fresh)	**Al**: flavouring in several dishes, ingredients in the traditional *mocetta*. **Dom/Hand**: macerate of the bulbs used spread on plants and animals to keep insects away. **Med**: as vermifuge: bulbs juice (drop ingestion), necklace of garlic, bulbils to spread on the belly with oil (**cip**); chewed garlic to promote the regularity of blood pressure (**circ**). **Vet**: choleretic infusion and cholagogue for dogs.	**Al**: [5,10,13] **Med**: [5,7,10,11,13] (**cip**)	0.44
*Allium schoenoprasum* L. A.sch.HBPNGP_ETN	Brenlette/Branlette (C)	W	Leaves (fresh)	**Al**: flavouring in omelettes. **Vet**: decoction as emollient, to be given to cows before the birth of calves.	**Al**: [5,6,7,10,13]	0.02
**Apiaceae**						
*Angelica sylvestris* L. A.syl.HBPNGP_ETN	Angelica	W	Stems (fresh or dried)	**Dom/Hand**: hollow stems used as whistles.		0.06
*Bunium bulbocastanum* L. B.bul.HBPNGP_ETN	Les adissons/Erdeusson (C) Tsemòtte (V)	W	Tubers (fresh)	**Al**: fresh tubers eaten raw as a chestnut-flavored snack.	[5,6]	0.25
*Carum carvi* L. C.car.HBPNGP_ETN	Cumino dei prati Kummel/Kumel (C-V-R)	W	Seeds (fresh or dried)	**Al**: flavouring several dishes. **Liq**: seeds flavouring in liqueur (called Kummel) and grappa as digestive. **Med**: infusion to drink as carminative, digestive and against gastric pains (**dig**). **Vet**: seeds placed in a butter ball and given to cows against difficulty in ruminating and swelling.	**Al**: [7,9,10,11] **Liq**: [5,7,9,10,11] **Med**: [5,7,8,9,10,11] (**dig**)	0.56
*Daucus carota* L.	Carota Rè-de-gneuf (C)	C	Roots (fresh)	**Al**: eaten raw or cooked, ingredient in the traditional *repùta*.	[7]	0.03
*Foeniculum vulgare* Mill.	Finocchietto selvatico Fenoglie (C) Feneuill (C)	P	Seeds (dried)	**Med**: seeds infusion against digestive problems and swelling (**dig**).	[13] (**dig**)	0.02
*Heracleum sphondylium* L. H.sph.HBPNGP_ETN	Tsaramé (C) Tsàamì (R)	W	Stems (fresh) Leaves (fresh)	**Dom/Hand**: hollow stems used as whistles; hollow stems used to inflate the bladder of the calf to be dried as rennet. **For**: as fodder for rabbits.	**For**: [5,7,10,11]	0.03
*Levisticum officinale* W.D.J. Koch L.off.HBPNGP_ETN	Sedano di montagna Celerì suvàdzu (C) Opio/Apio montano (R)	C/W	Leaves (fresh) Stems (fresh)	**Al**: ingredient in soups, flavouring in several dishes. **Vet**: leaves rubbed on mules, donkeys and horses as an insecticide, against flies and horseflies.		0.15
*Petroselinum crispum* (Mill.) Fuss	Prezzemolo Persì (C)	C	Leaves (fresh)	**Al**: flavouring in several dishes. **Med**: ingestion of large doses of leaves was used to cause abortion (**pcp**).	**Al**: [10,13] **Med**: [10,13] (**pcp**)	0.03
*Peucedanum ostruthium* (L.) W.D.J.Koch syn *Imperatoria ostruthium* L. P.ost.HBPNGP_ETN	Imperatoria Agrù (C-V-R) Agroù (C)	W	Rhyzomes Leaves Flowers (fresh or dried)	**Dom/Hand**: roots fumigations as disinfectants for the stables. **Liq**: roots flavouring in liqueur and grappa as digestive. **Med**: to treat skin problems: leaves, sometimes sprinkled of hot oil or butter, directly applied; leaf or root decoction used as compress (e.g., against wounds, burns, thorns, insect bites, infections, etc.) (**skin**) (**cip**); compress against muscle inflammation and contusions, hematomas and rheumatic pains; decoction for foot baths and compresses against leg and knees pain (**musc**); roots or leaves directly applied against caries or mouth ulcers and abscesses (**dent**); decoction for vaginal washings in case of infections or after delivery (**uro-gen**) (**pcp**); ointment of chopped roots mixed with marmot fat applied to the chest against respiratory problems (**resp**); flowers infusion to be ingested against inflammations and fever (**abn**); chopped root added in the beaten egg yolk and ingested as invigorator (**enm**). **Vet**: chopped root placed in a butter ball given to livestock against digestive problems; decoction or minced root mixed with fat or butter used to treat hoof problems; leaves and roots decoction used as external and internal disinfectant post-partum for cows.	**Dom/Hand**: [15] **Med**: [10,15] (**dent**) **Vet**: [5,15]	0.97
*Pimpinella anisum* L. P.ani.HBPNGP_ETN	Anice Anis (C) Feunòglie/Fenùlle (C-V-R)	W	Seeds (fresh or dried)	**Liq**: Flavouring in liqueur and grappa as digestive.	[10,11]	0.07
**Araliaceae**						
*Hedera helix* L. H.hel.HBPNGP_ETN	Edera maschio	W	Leaves (fresh)	**Med**: poultice of leaves mixed with honey ingested as vermifuge (**cip**).		0.03
**Aristolochiaceae**						
*Aristolochia clematitis* L. A.cle.HBPNGP_ETN	Aristolochia	W	Leaves (fresh) Flowers (fresh)	**Med**: leaves or flowers rubbed on infected pimples (**skin**).		0.02
**Asparagaceae**						
*Paradisea liliastrum* (L.) Bertol. P.lil.HBPNGP_ETN	Giglio bianco Paradisea Fieur de Lys (C)	W	Flowers (fresh or dried)	**Others**: flowering stems used to build a cross placed on the doors on the feast of St. John against misfortune.		0.06
*Ruscus aculeatus* L. R.acu.HBPNGP_ETN	Pungitopo	P	Branches (fresh or dried)	**Dom/Hand/Vet**: plant kept in stables as disinfectant, indicated against cow skin diseases such as *les dardes*.		0.03
**Asteraceae**						
*Achillea erba-rotta* All. A.erb.HBPNGP_ETN	Erba-rota Fernèt (C-V-R) Fèarnet (C) Fleur blanc (V-R)	W	Aerial parts (fresh or dried)	**Liq**: Flavouring in liqueur and grappa as digestive. **Med**: infusion in water or milk, with honey, against cooling and fever; vapors (to be inhale) against respiratory system diseases (**resp**) (**abn**); infusion against digestive (**dig**), urinary (**uri**) and circulatory (**circ**) problems. **Others**: harvested plants sold to liquor industry as source of income.	**Liq**: [5,6,7,10,11] **Med**: [5,7,10,11] (**resp**) [5,6,7,9,10,13] (**dig**) [9,10] (**uri**)	0.71
*Achillea moschata* Wulfen A.mos.HBPNGP_ETN	Fernèt (C-V-R) Fleur blanc (V-R)	W	Aerial parts (fresh or dried)	As above (see *A. erba-rotta*).	[12] (**dig**)	0.06
*Achillea millefolium* L. A.mil.HBPNGP_ETN	Achillea Millefoglie Moufètte (C) Fleur blanc (V-R)	W	Leaves (fresh) Flowers (dried)	**Liq**: flowers flavouring in grappa as digestive. **Med**: flowers infusion (to drink) against digestive (**dig**) and urinary (**uri**) problems; fresh leaves placed on wounds and burns as haemostatic and cicatrizing; leaves applied directly to the skin (sometimes after heating in butter) or leaves decoction (compress) on hematomas (**skin**) (**musc**). **Others**: dried leaves as tobacco substitutes. **Vet**: infusion against cows’ cough, colic and digestion disorders.	**Liq**: [10,11] **Med**: [5,8,9,10,11] (**dig**) [9,10] (**uri**) [5,9,10,11,12] (**skin**) **Vet**: [9,10,11]	0.28
*Arctium lappa* L. A.lap.HBPNGP_ETN	Bardana Les dzògne/Dzògnes (C) Gliògnes (R)	W	Leaves (fresh) Roots (fresh) Fruits (fresh)	**Al**: roots eaten raw. **Med**: leaves used as a compress (sometimes smeared with oil or hot butter) placed on wounds (antihemorrhagic), on rheumatic areas, on areas of arthrosis and arthritis, in the case of gout, on bruises, on aching back and knees, and to extract water from knees; root decoction (compress) on irritated skin or against acne (**skin**) (**musc**) (**circ**). **Others**: prickly heads thrown on clothes by children. **Vet**: leaf or root decoction as a diuretic for horses; used against bladder problems.	**Al**: [10] **Med**: [9,10,11,15] (**circ**) (**skin**) [6] (**dig**) **Others**: [13]	0.25
*Arnica montana* L. A.mon.HBPNGP_ETN	Arnica Ànica (C)	W	Flowers (fresh or dried) Leaves (dried)	**Med**: flower juice (drops), flower decoction (wrap or footbaths), flower macerate in oil or alcohol rubbed on rheumatic areas, hematomas, muscle pains, cramps, chilblains, swelling, breast swelling and sprains (**musc**) (**circ**); against sun burns (**skin**); digestive infusion (**dig**). **Others**: dried leaves as tobacco substitutes; plants harvested to be sold, source of income. **Vet**: flower decoction (compress) on swellings and swollen breasts.	**Med**: [5,6,7,8,9,10,11,13,16] (**musc**) [5] (**skin**) [5,10,16] (**dig**) **Others**: [16]	0.78
*Artemisia absinthium* L. A.abs.HBPNGP_ETN	Assenzio Ensein/Ensèn (C)	W	Leaves and Flowers (fresh or dried)	**Dom/Hand**: in closets against moths, in the cellar against mice; fumigations to disinfect the environment (mainly after the birth of calves); macerate as insecticide for plants (combined with nettle). **Liq**: flavoring in liqueur as digestive. **Med**: flowers ingestion (a teaspoon with honey) or infusion (to drink or to inhale) to stimulate appetite, against ulcer and stomach problems (**dig**) (**enm**); against cough and cold (**resp**); as anthelmintic (also massaged on the stomach or placed under the pillow) (**cip**); at high doses as abortifacient (**pcp**); lying on a bed of wormwood in case of rheumatic pain, arthrosis, growing pains; compress against circulatory problems (**circ**) (**musc**); warmed and placed behind the ears against ear pain (**ear**). **Others**: harvested plants sold as source of income. **Vet**: fresh leaf poultice mixed with pork fat to treat cows’ hooves; given to swollen cows to restart rumen; as bed for calves with growth problems.	**Dom/Hand**: [5,11] **Liq**: [6] **Med**: [5,6,7,9,10,13] (**dig**) [10,11] (**resp**) [5,6,7,9,10,13] (**cip**) [10] (**musc**) [10] (**dent**) **Others**: [7,10,13]	0.72
*Artemisia campestris* L. subsp. *borealis* (Pall.) H.M.Hall and Clem. A.cam.HBPNGP_ETN	Artemisia	W	Aerial parts (fresh or dried)	**Liq**: flavouring in liqueur and grappa as digestive.		0.04
*Artemisia glacialis* L. A.gla.HBPNGP_ETN	Genepì (C) (V) (R) Zenepì (C)	W	Aerial parts (fresh or dried)	**Liq**: flavouring in liqueur and grappa as digestive.		0.03
*Artemisia genipi* Stechm. A.gen.HBPNGP_ETN	Genepì (C) (V) (R) Zenepì/Zenepì màtchou (C) Genepì maschio	W	Aerial parts (fresh or dried)	**Al**: ingredient for the preparation of rennet to make cheese (maceration in water with calf bladder and *Galium*) **Liq**: flavouring in liqueur and grappa as digestive. **Med**: infusion against digestive problems (**dig**); infusion in water or milk with honey, vapors to inhale against cooling, fever, and respiratory system’ diseases (**resp**) (**abn**) (*). **Others**: harvested plants sold to liquor industry as source of income.	**Liq**: [5,7,9,10,13] **Med**: [5,9,10,11] (**dig**) [5,10] (**resp**) **Others**: [5]	0.74
*Artemisia umbelliformis* Lam. A.umb.HBPNGP_ETN	Genepì (C) (V) (R) Zenepì/Zenepì fèmmé (C) Mutellina Genepì bianco Genepì femmina	W	Aerial parts (fresh or dried)	As above (see *A. genipi*).		0.16
*Artemisia pontica* L. A.pon.HBPNGP_ETN	Assenzio gentile	W	Aerial parts (fresh or dried)	**Others**: harvested plants traded in the past as source of income.		0.02
*Artemisia vulgaris* L. A.vul.HBPNGP_ETN	Assenzio volgare	W	Aerial parts (fresh or dried)	**Med**: infusion against menstrual pains (**gen**).	[5,10] (**gen**)	0.02
*Calendula officinalis* L. C.off.HBPNGP_ETN	Calendula	C/W	Leaves and Flowers (fresh or dried)	**Med**: fresh leaves on wounds and burns; oil (flower macerate) on burns, or on dry, chapped, and inflamed skin (**skin**).	[9,10,11,13] (**skin**)	0.09
*Carduus defloratus* L. C.def.HBPNGP_ETN	Cardo Tsardòn	W	Leaves and Receptacles (fresh or dried)	**Al**: used to make curdled milk. **Med**: receptacles boiled and eaten as galactogogue (**pcp**). **Vet**: added to the mash of cows to increase milk production.		0.12
*Carlina acaulis* L. C.aca.HBPNGP_ETN	Cardo Tsardòn/Etzeardòn Plante du temps	W	Leaves and Receptacles (fresh or dried)	**Al**: boiled receptacles eaten as food. **Dom/Hand**: hanging on houses’ doors as a timepiece plant. **Med**: boiled and eaten receptacles as galactogogue (**pcp**); flowers infusion as digestive (**dig**). **Vet**: added in the mash of cows, increasing milk production.	**Al**: [5,7,9,10,13] **Dom/Hand**: [5]	0.16
*Centarea cyanus* L. C.cya.HBPNGP_ETN	Fiordaliso	C/W	Flowers (fresh or dried)	**Med**: soothing infusion against nervousness (**nerv**).		0.02
*Cirsium eriophorum* (L.) Scop. C.eri.HBPNGP_ETN	Cardo Etzeardòn	W	Receptacles (fresh or dried)	**Al**: edible receptacles. **For**: forage appreciated by donkeys.		0.03
*Doronicum grandiflorum* Lam. subsp. *grandiflorum* D.gra.HBPNGP_ETN	Sabadeille	W		Plant whose dialectal name is known.		0,02
*Leontopodium nivale* (Ten.) È. Huet and A.Huet ex Hand.-Mazz. subsp. *alpinum* (Cass.) Greuter L.niv.HBPNGP_ETN	Stella alpina	W	Flowers (fresh or dried)	**Dom/Hand**: used as bookmark or ornament.	[11]	0.10
*Matricaria chamomilla* L.	Camomilla	P	Flowers (dried)	As below (see *M. discoidea*).		0.22
*Matricaria discoidea* DC M.dis.HBPNGP_ETN	Camomilla selvatica	W	Flowers (dried)	**Med**: infusion (to drink) as soothing, sedative, against head pain (**nerv**), as digestive and against gastrointestinal pain (**dig**); flowers warmed in oil and put on ear lobes in case of earache (**ear**); infusion (compress) against eye inflammation and infections (sometimes mixed to beaten egg white) (**eye**) (**cip**); wraps on infected wounds (**skin**); chamomile baths for agitated children (**pcp**); footbaths in chamomile infusion for tired and blistered feet (**musc**).	[8,9,10,11,12,13] (**nerv**) [10] (**dig**) [7,10,13] (**eye**) (**ear**) [10] (**skin**)	0.32
*Petasites hybridus* (L.) G. Gaertn., B.Mey. and Scherb. P.hyb.HBPNGP_ETN	Farfaraccio	W	Leaves (fresh)	**Med**: leaves placed on bruises (**musc**).	[9] (**skin**)	0.02
*Tanacetum vulgare* L. T.vul.HBPNGP_ETN	Tanaceto Fiori della Madonna Archebùse/Archebùe(C) Boutòn du vers	C	Flowers and Leaves (fresh)	**Liq**: leaves flavoring in liqueur (called arquébuse) and grappa as digestive. **Med**: flower infusion, in water or milk, or intake of crumbled flowers (a teaspoon) with honey as vermifuge (**cip**).	**Med**: [5,7,10] (**cip**)	0.25
*Taraxacum officinale* aggr. T.off.HBPNGP_ETN	Tarassaco Cicoria Cicorie Tzecòrie/Zeucorie (C)	W	Leaves and Flowers (fresh) Roots (dried) Floral Stems (fresh or dried)	**Al**: young leaves as an ingredient in salad with boiled eggs, potatoes and walnut oil; the oldest leaves in soups and omelet or stir-fried with butter; buds in brine or pickle, substitutes for capers; roasted and powdered roots as a coffee substitute. **For**: excellent fodder for hens and rabbits. **Med**: leaf infusion or root decoction as a liver purifier (**dig**). flower syrup (honey substitute) against cough and as depurative (**resp**). **Others**: hollow floral stems used to build whistles.	**Al**: [5,6,7,9,10,11] **For**: [7] **Med**: [7,12] (**uri**) [11] (**dig**) [9,10] (**resp**) **Others**: [6,11]	0.81
*Tragopogon pratensis* L. T.pra.HBPNGP_ETN	Erba bòch/Erba bec (C), Barba boc (R)	W	Leaves and Floral Stems (fresh) Roots (fresh or dried)	**Al**: floral stems eaten as a snack; leaves in soups and omelet or stir-fried with butter; roots in soups. **Med**: leaf decoction as depurative and diuretic (**uri**).	**Al**: [5,6,7,10,11]	0.40
*Tussilago farfara* L. T.far.HBPNGP_ETN	Farfaró Fàrfaro Fàrfara	W	Flowers (fresh or dried) Leaves (fresh)	**Med**: infusion or decoction, in water or milk, (to drink) against cough, bronchitis, phlegm, and as expectorant; infusion (to inhale) against respiratory problems and fever; antitussive syrup (**resp**) (**abn**) (*); flower infusion as digestive (**dig**); leaves placed on wounds (**skin**). Attention high doses can cause health problems, recommended dose 4–5 flowers per cup.	[5,7,8,9,10,11,13,16] (**resp**) [10,13,16] (**skin**)	0.60
**Berberidaceae**						
*Berberis vulgaris* L. B.vul.HBPNGP_ETN	Crespino Berberis Les Pàppes (C)	W	Fruits (fresh) Branches (fresh or dried)	**Al**: ripe fruits (collected after the first frost) eaten raw or lightly toasted on the stove; ingredient in syrups and jams; macerated to make vinegar; fruits grounded to make laxative flour. **Dom/Hand**: branches used to build flat brooms for cleaning grain and hay in the barn; spiny branches placed on sown areas in vegetable gardens to block access to hens. **For**: fruits to feed goats and pigs. **Liq**: fruit macerate as ingredient for the Cogne beer recipe (with *Hordeum vulgare* and *Polypodium vulgare*); fruits soaked in water and sugar to make wine. **Med**: fruit decoction to drink as digestive (**dig**).	**Al**: [7,10,11,16] **Med**: [10] (**dig**)	0.77
**Betulaceae**						
*Alnus viridis* (Chaix) DC A.vir.HBPNGP_ETN	Ontano Toùsa (R) Drouse (C)	W	Plant	**Others**: water purifier denoting the good quality of water.		0.02
*Betula pendula* Roth B.pen.HBPNGP_ETN	Betulla Bioùla (C)	W	Leaves (fresh) Lymph (fresh) Cortex (fresh) Branches (dried)	**Dom/Hand**: branches used to build brooms for cleaning stables; bark used to build snuffboxes. **Med**: leaf infusion (to drink) as diuretic and purifier, against circulatory problems and rheumatism; lymph extracted in spring, (to drink) as purifier, as invigorating and mineralizing (**uri**) (**circ**) (**dig**) (**enm**); bark placed on insect or viper bites (**skin**).	**Med**: [10,15] (**uri**)	0.24
*Corylus avellana* L.	Nocciolo	P	Fruits (fresh) Branches (fresh)	**Al**: edible fruits. **Dom/Hand**: young branches used to build baskets.		0.02
**Boraginaceae**						
*Borago officinalis* L.	Borraggine	C	Leaves (fresh)	**Al**: young leaves in soups and omelet or stir-fried with butter.	[6,7,9,10,11,13]	0.09
*Myosotis alpestris* F.W. Schmidt M.alp.HBPNGP_ETN	Miosotì	W	Flowers (fresh)	**Dom/Hand**: flowers used as decoration (long survivals).	[11]	0.02
**Brassicaceae**						
*Brassica nigra* (L.) W.D.J. Koch	Senape	P	Seeds (fresh)	**Vet**: mustard smeared on hard and chronic swellings in order to exacerbate and treat them.		0.02
*Brassica oleracea* L.	Cavolo Tzùt (C)	C	Leaves (fresh)	**Al**: leaves as ingredient in several recipes (e.g., in the *repùta* recipe). **Med**: leaf compress on aching knees, to extract water from knees (**circ**) (**musc**).	**Al**: [5,13] **Med**: [11] (**circ**) (**musc**)	0.27
*Brassica rapa* L.	Rapa Ràve Bètterave/Bèllerave/Bèterava	C	Leaves (fresh) Roots (fresh)	**Al**: leaves and roots as ingredients in several recipes (e.g., in the *repùta* recipe). **For /Vet**: given to the cows as galactagogue.	**For/Vet**: [10]	0.25
*Brassica rapa* L. subsp. *rapa*	Rapa bianca Ràve	C		**Med**: antitussive syrup against respiratory problems (cutting roots into slice and sprinkling them with sugar) (**resp**).	[7] (**resp**)	0.16
*Capsella bursa-pastoris* (L.) Medik C.bur.HBPNGP_ETN	Borsa pastore Porta Pan (C)	W	Flowers and Seeds (dried)	**Med**: infusion (to drink) as diuretic, against urinary diseases, and kidney stones (**uri**).		0.02
**Campanulaceae**						
*Phyteuma ovatum* Honck. P.ova.HBPNGP_ETN	Bon-hòmmo (C) Gros hòmmos (C)	W	Leaves (fresh)	**Al**: ingredient in soups and omelettes.	[6,7,11,16]	0.09
**Cannabaceae**						
*Cannabis sativa* L.	Canapa Oneisse (R)	P	Fibers (fresh and dried)	**Dom/Hand**: fiber used for weaving.	[13]	0.03
*Humulus lupulus* L.	Luppolo	P	Flowers (fresh or dried)	**Liq**: ingredient in beer.		0.02
**Caprifoliaceae**						
*Lonicera caerulea* L. subsp. *caerulea* L.cae.HBPNGP_ETN	Cobbie	W		Plant whose dialectal name is known.		0.02
*Valeriana celtica* L. subsp. *celtica* V.cel.HBPNGP_ETN	Aspèch (C)	W	Aerial parts (fresh or dried), Roots (fresh or dried)	**Dom/Hand**: placed in closets and drawers against moths. **Med**: chopped root to calm headache (**nerv**) and bellyache (**dig**) (1 teaspoon).		0.22
*Valeriana officinalis* L.	Valeriana	P	Roots (fresh or dried)	**Med**: shooting infusion, against headache (**nerv**).	[10,13] (**nerv**)	0.09
**Caryophyllaceae**						
*Agrostemma githago* L. A.git.HBPNGP_ETN	Neglie	W		Plant whose dialectal name is known.		0.02
*Silene vulgaris* (Moench) Garcke S.vul.HBPNGP_ETN	Silene Tic e tac/Tchatchac (C) Fleur de pentecouta (C) Puf (R) Fleur du tac (V)	W	Leaves (fresh)	**Al**: ingredient in soups and omelettes or stir-fried with butter. **Vet**: given to the cows to restore rumen.	**Al**: [6,7,9,10,11,13]	0.28
**Colchicaceae**						
*Colchicum autumnale* L. C.aut.HBPNGP_ETN	Pourès (C) Pourètte (R) Boètte (R)	W	Seeds (dried)	**Others**: capsules containing seeds collected and sold to pharmaceutical companies as source of income.	[13]	0.04
*Colchicum bulbocodium* Ker Gawl. C.bul.HBPNGP_ETN	Poures (C)	W		Plant whose dialectal name is known.		0.02
**Convolvulaceae**						
*Convolvulus arvensis* L. C.arv.HBPNGP_ETN	Corioule	W		Plant whose dialectal name is known.		0.02
**Cornaceae**						
*Cornus sanguinea* L. C.san.HBPNGP_ETN	Sanguinello	W	Fruits (fresh)	**Al**: edible black berries (not appreciated).		0.02
**Crassulaceae**						
*Hylotelephium maximum* (L.) Holub syn *Sedum telephium* L. H.max.HBPNGP_ETN	Erba di tutti i mali /Foglia de chiumaux (V)	C/W	Leaves (fresh)	**Med**: leaf poultice spread on rheumatic areas, infected wounds, and thorns to facilitate extraction (**musc**) (**circ**) (**skin**).	[10] (**skin**)	0.07
*Sedum album* L. S.alb.HBPNGP_ETN	Piquet d’ésoui (C) Pequè di ci-cich (C) Piquin	W	Leaves (fresh)	**Med**: leaf poultice to treat skin diseases (**skin**)		0.04
**Cucurbitaceae**						
*Cucurbita maxima* Duchesne	Zucca	C	Seeds (fresh)	**Med**: raw or roasted seeds (to ingest) as a vermifuge (**cip**).	[13] (**cip**)	0.03
**Cupressaceae**						
*Juniperus communis* L. J.com.HBPNGP_ETN	Ginepro Tsenévru (C) Tsénèvro Tsenevrò	W	Fruits (fresh or dried) Branches (fresh or dried) Wood (dried)	**Al**: flavouring in several dishes (e.g., meat, game, *repùta*, *vin brulé*, cheeses). **Dom/Hand**: branches fumigations to disinfect environments; large branches used to build the stick (*moudòn, moudèire*) used to turn the polenta, releasing an aromatic flavor. **Liq**: flavouring in liqueur and grappa as digestive. **Med**: fresh berries, or berry infusion or decoction (in water or wine) against stomach ache, digestive problems, indigestion; berry jam (called *tsénèvrà*) used to treat digestive (**dig**), respiratory (**resp**), circulatory (**circ**), urinary problems (**uri**), against osteoarthritis and osteoporosis (**skel**), and in case of tooth pain (**dent**), against headache (**nerv**) and menstrual pain (**gen**). **Vet**: berries added to balls of fat or butter, or in form of decoction given to animals with rumen problems and against swelling.	**Al**: [5,6,7,9,10,11,13,15] **Dom/Hand**: [9,11,13] **Liq**: [5,6,8,9,10,13] **Med**: [7,10,11,15] (**dig**) [9,10,11,13,15] (**resp**) **Vet**: [7]	0.88
*Juniperus sabina* L. J.sab.HBPNGP_ETN	Sabina Silèn (C)	W	Fruits (fresh)	**Med**: berry intake used to cause abortion (**pcp**).	[10] (**pcp**)	0.02
**Elaeagnaceae**						
*Hippophae rhamnoides* L. H.rha.HBPNGP_ETN	Olivello spinoso Engòsse (C)	W	Fruits (fresh) Branches (fresh or dried)	**Al**: fruits eaten raw, as ingredient in jams. **Dom/Hand**: spiny branches put on sown areas in vegetable gardens to block access to hens. **Med**: jam as invigorating and preventive for skin (**skin**) and respiratory problems (**resp**).	**Al**: [10,11] **Med**: [10]	0.16
**Equisetaceae**						
*Equisetum arvense* L. E.arv.HBPNGP_ETN	Equiseto Erba cavallina Èrbe mouròn (C) L’écuille (V)	W	Steril stems (fresh or dried)	**Med**: dried and powdered (to ingest) or as infusion or decoction for weak bones, against osteoporosis, arthritis, arthrosis, in case of rheumatism, kidney or urinary problems, to regularize the menstrual cycle, as invigorator and reinforcer (**skel**) (**enm**) (**gen-uri**). **Others**: segments were used by children as a game, because once separated they reassemble to each other.	**Med**: [9,10,11] (**skel**) [6,7,9,10,11,15] (**uri**)	0.18
**Ericaceae**						
*Arctous alpina* (L.) Nied. A.alp.HBPNGP_ETN	Embeuretze Embrùtzo/Embreutso (C)	W	Fruits (fresh)	**Al**: fruits eaten raw.		0.07
*Arctostaphylos uva-ursi* (L.) Spreng. A.uva.HBPNGP_ETN	Uva ursina Fareneule/Farenna èn-soula/Farenùla (C) Farenoula (R) Gramòn (R) Reiselèn (V)	W	Fruits (fresh) Leaves (fresh or dried)	**Al**: edible fruits. **Med**: leaf decoction against inflammations and infections of the urinary tract, cystitis and prostatitis (**gen-uri**). **For/Vet**: fruit flour as forage for calves; leaf decoction against inflammations and infections of the urinary system.	**Al**: [7,10] **Med**: [7,10,11] (**uri**) [10] (**gen**)	0.32
*Rhododendron ferrugineum* L. R.fer.HBPNGP_ETN	Rododendro Frainbitchou/Frembitcho (C)	W	Flowers (fresh) Branches (fresh or dried)	**Dom/Hand**: branches used on sown areas in vegetable gardens to block access to hens. **Others**: good melliferous plant.	**Dom/Hand**: [13]	0.03
*Vaccinium myrtillus* L. V.myr.HBPNGP_ETN	Mirtillo Lioùtre (C) Loùffie (R e V)	W	Fruits (fresh) Leaves (fresh)	**Al**: fruits eaten raw or as ingredient in jams. **Liq**: flavouring in grappa. **Med**: fruit and leaf infusion or decoction (to ingest) as antidiarrheal (**dig**), against circulatory problems (**circ**), urinary tract problems, and prostatitis (**gen-uri**); decoction (compress) on inflamed eyes (**eye**).	**Al**: [5,6,7,9,10,11,13] **Liq**: [6] **Med**: [9,10,11] (**dig**) (**circ**) [10,11] (**uri**) (**eye**)	0.85
*Vaccinium uliginosum* L. V.uli.HBPNGP_ETN	Falso mirtillo	W	Fruits (fresh)	**Al**: fruits eaten raw.	[5,10,11]	0.04
*Vaccinium vitis-idaea* L. V.vit.HBPNGP_ETN	Mirtillo rosso Gravelòn/Graveulon (C)	W	Fruits (fresh) Leaves (fresh)	**Al**: fruits eaten raw, dried and stored in glass jars, or as ingredient in jams. **Med**: leaf decoction (to drink) against urinary tract problems and prostatitis (**gen-uri**).	**Al**: [5,7,9,10,11] **Med**: [9,11] (**uri**)	0.31
**Euphorbiaceae**						
*Euphorbia seguieriana* Neck E.seg.HBPNGP_ETN	Fleur du serpent	W	Latex (fresh)	**Med**: toxic latex put on warts and infected skin (e.g., fungi) (**skin**).	[10]	0.06
*Euphorbia helioscopia* L. E.hel.HBPNGP_ETN	Euforbia	W	Latex (fresh)	As above (see *E. seguieriana*).		0.02
**Fabaceae**						
*Astragalus alopecurus* Pall.	Coda di volpe	W	Flowers (dried)	**Dom/Hand**: dried flowers as decoration.		0.02
*Lotus corniculatus* L. L.cor.HBPNGP_ETN	Loto	W	Flowers and Leaves (fresh or dried)	**Al**: ingredient for cheese curdle, releasing a yellowish color.		0.02
*Medicago sativa* L. M.sat.HBPNGP_ETN	Erba medica Sanfouèn (C)	C/W	Leaves and Flowers (fresh or dried)	**For**: excellent forage for cows, galactogogue.	[7,10,11,13]	0.15
*Onobrychis viciifolia* Scop. O.vic.HBPNGP_ETN	Tsavretta Tsevrètta (C)	C/W	Leaves and Flowers (fresh or dried)	**For**: excellent forage for cows, galactogogue.	[5,7,13]	0.03
*Phaseolus vulgaris* L.	Fagiolo Fisoùs (C)	C	Seeds and Pods cuticles (fresh or dried)	**Al**: edible seeds. **For**: cuticles given to cows before giving birth.		0.04
*Trifolium alpinum* L. T.alp.HBPNGP_ETN	Trifoglio di montagna Seutrin (C) Trioùla (C) Sanfuèn (R)	W	Leaves and Flowers (fresh)	**Al**: flowers as ingredient in soups. **For**: excellent galactogogue fodder for cows, makes milk fatter and fontina cheese delicious.	**For**: [7]	0.24
*Trifolium pratense* L. T.pra.HBPNGP_ETN	Trifoglio dei prati Trioulette (C)	W	Leaves and Flowers (fresh)	**Al**: flowers and leaves as ingredients in soups sweetish flowers sucked as a snack. **Med**: flowers infusion against menopause symptoms and disorders (**enm**). **For/Vet**: excellent galactogogue forage for rabbits and cows.	**Al**: [7,10,11,13,16] **For/Vet**: [10]	0.21
*Trifolium repens* L. T.rep.HBPNGP_ETN	Trifoglio dei prati Trioulette (C)	W	Leaves and Flowers (fresh)	As above (see *T. pratense*).		0.02
*Vicia faba* L.	Fava Fave (C)	C	Seeds and Pods cuticles (fresh or dried)	**Al**: edible seeds. **For**: cuticles given to cows before giving birth.	**Al**: [5]	0.07
**Gentianaceae**						
*Gentiana acaulis* L. G.aca.HBPNGP_ETN	Genzianella Gentièn (C) Peirette (C)	W	Flowers (fresh or dried)	**Liq**: flowers macerate in wine as invigorating; flavouring in liqueur and grappa as digestive. **Med**: flowers infusion against digestive problems and as appetizer (**dig**), as invigorating (**enm**) against headache caused by cold and respiratory problems (**resp**).	**Liq**: [5,6,7,10,11] **Med**: [5,7,8,10,11] (**dig**)	0.22
*Gentiana verna* L. G.ver.HBPNGP_ETN	Genzianella Gentièn (C) Fieur di Corbas (C)	W	Flowers (fresh or dried)	As above (see *G. verna*).		0.10
*Gentiana lutea* L. G.lut.HBPNGP_ETN	Genziana Gentiànne Dzentsànna	W	Roots (fresh or dried)	**Liq**: flavouring in liqueur and grappa as digestive. **Dom/Hand**: roots boiled in water with ash for laundry. **Med**: root decoction against digestive and liver problems, as tonic and appetizer (**dig**); root decoction in wine as invigorating and against anemia (**circ**) (**abn**) (**enm**). Attention not to be confused with the similar species *Veratrum album* L.	**Liq**: [5,6,7,8,10,11,13] **Med**: [5,6,7,10,11,12,13] (**dig**) [5,10] (**circ**) [9,10] (**uri**)	0.24
*Gentiana punctata* L. G.pun.HBPNGP_ETN	Genziana Gentiànne Dzentsànna	W	Roots (fresh or dried)	As above (see *G. lutea*).		0.56
**Geraniaceae**						
*Geranium robertianum* L. G.rob.HBPNGP_ETN	Erba roberta Sepàh (R)	W	Leaves (fresh)	**Med**: leaf poultice, mixed with honey, smeared on hematomas (**musc**).		0.07
*Pelargonium* sp.	Geraniòn (C)	C	Leaves (fresh)	**Med**: fresh leaf poultice on insect bites (**skin**).	[7] (**skin**)	0.03
**Grossulariaceae**						
*Ribes alpinum* L. R.alp.HBPNGP_ETN	Ribes	W	Fruits (fresh)	**Al**: fruits eaten raw or as ingredient in jams.	[7,10]	0.07
*Ribes nigrum* L.	Ribes nero Cassì	C	Fruits (fresh)	**Al**: fruits eaten raw or as ingredient in jams or syrups.		0.18
*Ribes petraeum* Wulfen R.pet.HBPNGP_ETN	Ribes	W	Fruits (fresh)	**Al**: fruits eaten raw or as ingredient in jams.	[10,13]	0.07
*Ribes rubrum* L.	Ribes rosso Resin a-bràn (C)	C	Fruits (fresh)	**Al**: fruits eaten raw or as ingredient in jams.	[11,13]	0.37
*Ribes uva-crispa* L. R.uva.HBPNGP_ETN	Uva spina Grousèlles (C)	W	Fruits (fresh)	**Al**: fruits eaten raw or as ingredient in jams.	[7,13]	0.47
**Hypericaceae**						
*Hypericum perforatum* L. H.per.HBPNGP_ETN	Iperico	W	Flowers (fresh or dried)	**Med**: ointment made of flowers macerate in olive or linen oil to spread on burns, on sunburn, on eczema, skin erythema (**skin**), and aching nerves (**nerv**); flower infusion against digestive problems (**dig**).	[5,6,7,8,10,11,12,13] (**skin**) [10,11] (**nerv**) (**dig**)	0.10
**Iridaceae**						
*Crocus vernus* (L.) Hill C.ver.HBPNGP_ETN	Falso zafferano Catagnùla (C)	W	Pistils (fresh or dried)	**Al**: flavoring and coloring several dishes, especially risotto.	[10]	0.03
**Juglandaceae**						
*Juglans regia* L.	Noce Noyér (C)	P	Fruits (fresh or dried) Limph (fresh)	**Al**: edible fruits, fruits used to make oil. **Med**: walnut oil spread on blows and on hematomas (**musc**); walnut oil or lymph (to ingest) against colitis (**dig**).	**Al**: [5,6,10,13]	0.06
**Juncaceae**						
*Juncus jacquinii* L. J.jac.HBPNGP_ETN	Camousseire	W		Plant whose dialectal name is known.		0,02
**Lamiaceae**						
*Hyssopus officinalis* L.	Issopo	P	Flowers (fresh or dried)	**Others**: flowers collected and sold as source of income.		0.02
*Lavandula angustifolia* Miller	Lavanda	P	Flowers (fresh or dried)	**Dom/Hand**: perfumed sachets of flowers to put in closets and drawers. **Med**: flower infusion as a sedative, against insomnia and nervousness (**nerv**).	**Dom/Hand**: [11,13] **Med**: [10,11,13] (**nerv**)	0.06
*Melissa officinalis* L.	Melissa	P	Leaves and Flowers (fresh or dried)	**Med**: soothing, relaxing (**nerv**), and digestive (**dig**) infusion.	[9,10] (**nerv**) [10] (**dig**)	0.15
*Mentha longifolia* L. M.lon.HBPNGP_ETN	Menta	C/W	Leaves (fresh or dried)	**Al**: refreshing infusion. **Dom/Hand**: leaves placed on potatoes in the cellar to keep mice away. **Med**: soothing and relaxing infusion (**nerv**) and as digestive (**dig**) and diuretic (**gen-uri**); infusion (to inhale) against stuffy nose (**resp**); leaf pack on infected breast (**cip**); leaves rubbed on the breast to reduce milk production (**pcp**). **Vet**: boiled leaves as pack on infected udders and against mastitis.	**Med**: [5,10,11,13] (**nerv**) [7,8,9,10,11,13] (**dig**) [10] (**resp**)	0.22
*Nepeta cataria* L.	Cataria Erba du tsà (R) Earba di tset (C)	P	Leaves (fresh or dried)	**Med**: leaf infusion against headache (**nerv**).	[5,10] (**nerv**)	0.04
*Salvia officinalis* L. S.off.HBPNGP_ETN	Salvia selvatica Sèrve suvàdze (C)	P	Leaves (fresh or dried)	**Al**: aromatic; flavouring in several dishes (e.g., chamois in civet and *mocetta*). **Cosm**: leaves rubbed on teeth for dental hygiene. **Med**: relaxing infusion against fatigue (**nerv**), as digestive (**dig**), depurative (**uri**). leaf infusion in milk with honey, or in wine with honey and butter, as a febrifuge in case of colds or bronchitis (**resp**) (**abn**) (*); as compress on aching tooth (**dent**) or immature pustules (**skin**).	**Al**: [10,13] **Cosm**: [13] **Med**: [7,10,11] (**nerv**) (**dig**) (**dent**) [7] (**resp**)	0.49
*Salvia pratensis* L. S.pra.HBPNGP_ETN	Salvia dei prati Sèrve (C)	W	Leaves (fresh)	**Al**: ingredient in soups and omelettes.	[7]	0.03
*Salvia rosmarinus* Spenn. syn *Rosmarinus officinalis* L	Rosmarino Rusmarìn (C)	P	Leaves (fresh or dried)	**Al**: aromatic; flavouring in several dishes (e.g., chamois in civet and *mocetta*). **Med**: relaxing infusion against fatigue (**nerv**), as digestive (**dig**), depurative, antirheumatic (**uri**) (**circ**).	**Al**: [10,13] **Med**: [7,12] (**dig**)	0.10
*Satureja montana* L.	Santoreggia Parietta (R)	P	Leaves (fresh or dried)	**Al**: aromatic.	[11,13]	0.06
*Teucrium chamaedrys* L. T.cha.HBPNGP_ETN	Calamandrea	W	Aerial parts (fresh or dried)	**Med**: infusion against fever and colds (**resp**) (**abn**).	[16]	0.10
*Thymus pulegioides* L. syn *T. serpyllum* Auct. T.pul.HBPNGP_ETN	Timo serpillo Pouilloù (C) Tsarpolèt (V-R)	W	Aerial parts (fresh or dried)	**Al**: flavouring in several dishes (e.g., chamois in civet and *mocetta*). **Dom/Hand**: perfumed sachets of flowers in closets and drawers against moths. **Liq**: flavouring in liqueur. **Med**: flower infusion as a sedative, against insomnia and nervousness (**nerv**); infusion as digestive (**dig**) and vermifuge (**cip**); infusion, in water or milk with honey, against fever and colds (**resp**) (**abn**).	**Al**: [5,6,7,10,11,13,16] **Liq**: [7,10] **Med**: [5,6,7,8,9,10,11,13,16] (**dig**) [8,9,10,11,12,13,16] (**resp**)	0.47
**Lauraceae**						
*Laurus nobilis* L.	Alloro Luré (C)	P	Leaves (fresh or dried) Branches (fresh)	**Al**: flavouring in several dishes, especially meat (e.g., chamois in civet and *mocetta*). **Dom/Hand**: in closets and drawers against moths. **Med**: leaf infusion as digestive (**dig**). **Others**: branches decorated with an apple and sweets were blessed for Easter Sunday.	**Al**: [7,9,10,13] **Med**: [7,10,12,13] (**dig**)	0.16
**Lentibulariaceae**						
*Pinguicula* sp.	Pinguicola	W	Leaves (fresh)	**Med**: leaves used fresh or preserved in oil, put on wounds to help healing (**skin**).	[7,8,10,13] (**skin**)	0.02
**Liliaceae**						
*Lilium candidum* L.	Giglio bianco	C	Flowers (fresh)	**Med**: petals used fresh or preserved in oil or grappa, to spread on wounds and hematomas (**skin**) (**musc**).		0.09
**Linaceae**						
*Linum usitatissimum* L.	Lino Gran de lin	C	Seeds (fresh or dried)	**Med**: infusion, decoction, seeds macerated overnight in water or linen oil (to ingest) as an anti-inflammatory: against digestive problems, as a laxative but also antidiarrheal (**dig**) (**enm**), against respiratory problems, cough, cold, phlegm (also used as a chest pack) (**resp**); seed decoction used for bath against hemorrhoids (**circ**); seed decoction or water macerate (to drink) as a pre-birth emollient treatment (**pcp**). **Vet**: seeds decoction or macerate given to cows as a pre-birth emollient treatment.	**Med**: [5,7,9,10,11,13] (**resp**) [7,9,10,11,12,13] (**dig**) [10] (**pcp**) **Vet**: [10]	0.68
**Malvaceae**						
*Malva neglecta* Wallr. M.neg.HBPNGP_ETN	Malva Mèrve (C) Mèrva Marve	W	Aerial parts Leaves Flowers Seeds (fresh or dried)	**Al**: ingredient in soups. **Med**: relaxing and refreshing infusion (**nerv**); decoction against digestive (against abdominal pain and constipation) (**dig**) and respiratory system (**resp**) diseases; infusion or decoction (to drink) against cystitis and kidney problems (also as compress) (**uri**); infusion or decoction (to drink) as a pre-birth emollient treatment (**pcp**); decoction (external use) on inflamed areas: eye (**eye**), vaginal (**gen**) and dental (**dent**) washes, bath against hemorrhoids (**circ**); against internal infections (**cip**); leaves as a compress on inflamed areas, hematomas, and wounds (**musc**) (**skin**);	**Al**: [6,7,8,9,11,13,15] **Med**: [6,7,8,9,10,11] (**dig**) [8,10] (**resp**) (**gen**) [7,9,10,11,12] (**pcp**) (**uri**) [5,10,11,13] (**dent**) [10] (**skin**)	0.81
*Malva sylvestris* L. M.syl.HBPNGP_ETN	Malva Mèrve (C) Mèrva Marve	W	Aerial parts Leaves Flowers Seeds (fresh or dried)	As above (see *M. neglecta*).		0.09
*Tilia platyphyllos* Scop.	Tiglio	P	Flowers and Bracts (dried)	**Med**: relaxing and soothing infusion, against nervousness, sleepiness, and headache (**nerv**); infusion against fever and colds (**resp**) (**abn**).	[8,10] (**nerv**) [9,10,13] (**resp**)	0.16
**Melanthiaceae**						
*Veratrum album* L. V.alb.HBPNGP_ETN	Veratro Veratru (C) Valaio (C)	W	Leaves (fresh)	**Vet**: leaf macerate to spread on animals for keeping insects away.	[5,6,13]	0.02
**Myristicaceae**						
*Myristica fragrans* Houtt.	Noce moscata Nué de muscat (C)	P*	Seeds (dried)	**Al**: as aromatic. **Med**: nutmeg flavored coffee (to drink) against menstrual pain (**gen**) (**enm**).	**Med**: [7] (**gen**)	0.02
**Myrtaceae**						
*Syzygium aromaticum* (L.) Merr. and L.M.Perry	Chiodi di garofano Ciut de garòffie (C)	P*	Seeds (dried)	**Al**: aromatic and preservative, as ingredient in *repùta* and *vin brulé*.		0.03
**Oleaceae**						
*Fraxinus excelsior* L. F.exc.HBPNGP_ETN	Frassino Frèinu/Freino (C)	W	Wood (dried)	**Dom/Hand**: hard wood to build handles of pickaxes and shovels, and the board for cheese resting. Bark used to build whistles.	[13]	0.21
*Syringa vulgaris* L.	Lillà Lillò (C)	C	Flowers (fresh)	**Dom/Hand**: decorative and fragrant bunches.		0.02
*Olea europaea* L.	Ulivo	P	Fruits (fresh)	**Al**: oil used as seasoning. **Med**: oil heated and put on the ear lobe in case of earache (**ear**).	**Al**: [13] **Med**: [10,13] (**ear**)	
**Onagraceae**						
*Epilobium angustifolium* L. E.ang.HBPNGP_ETN	Epilobio Venturìn (C) Fieur de sent’Anne (C)	W	Flowers (fresh or dried)	**Med**: infusion against prostatitis and gout (**gen-uri**) (**circ**).		0.04
**Ophioglossaceae**						
*Botrychium lunaria* (L.) Sw. B.lun.HBPNGP_ETN	Lunaria Il Diavolo e la Madonna Bechet et Madona Bon Jeu- Bequet	W	Aerial Parts (fresh)	**Al**: ingredient in soups and omelettes or stir-fried with butter. **For**: forage appreciated by cows.		0.24
**Orchidaceae**						
*Gymnadenia nigra* (L.) Rchb. f. Syn *Nigritella nigra* (L.) Rchb f. G.nig.HBPNGP_ETN	Nigritella Tsencòn (C)	W	Aerial Parts (fresh)	**Al**: to make milk curdle. **Dom/Hand**: bunches smelling like vanilla and chocolate placed in houses or donated to damsels (aphrodisiac fragrance). **For**: if eaten in large amounts by cows fontina becomes bitter. **Liq**: flavoring in grappa. **Med**: infusion, in milk or water, (to drink or inhale) against respiratory system diseases, as a heater (**resp**) (**abn**) and as digestive (**dig**); aphrodisiac infusion.	**Dom/Hand**: [11] **Med**: [9,16]	0.37
**Orobanchaceae**						
*Euphrasia officinalis* L. subsp. *rostkoviana* (Hayne) Towns. E.off.HBPNGP_ETN	Eufrasia	W	Aerial Parts (fresh or dried)	**Med**: infusion (as compress) on inflamed eyes (**eye**).	[7,8,9,10,16] (**eye**)	0.06
*Rhinanthus alectorolophus* (Scop.) Pollich R.ale.HBPNGP_ETN	Tataneire (C)	W		Plant whose dialectal name is known.		0.02
**Oxalidaceae**						
*Oxalis acetosella* L. O.ace.HBPNGP_ETN	Acetosella	W	Leaves, Aerial Parts (fresh)	**Al**: leaves in salads; to make milk curdle.	[7,10,16]	0.06
Papaveraceae						
*Chelidonium majus* L. C.maj.HBPNGP_ETN	Chelidonia	W	Latex, Leaves (fresh)	**For**: leaves given to hens for increasing eggs production. **Med**: latex topically applied for treating warts (**skin**).	**Med**: [5,6,7,8,9,10,11,12,13] (**skin**)	0.15
*Papaver somniferum* L.	Oppio	C	Flowers and Capsules (fresh or dried)	**Med**: soothing infusion (**nerv**).		0.02
**Pinaceae**						
*Abies alba* Mill. A.alb.HBPNGP_ETN	Beuzon (C)	W		Plant whose dialectal name is known.		0.02
*Larix decidua* Mill. L.dec.HBPNGP_ETN	Larice Brenva (C)	W	Wood (dried) Needles (dried) Immature Cones (fresh) Resin (fresh or dried) Vascular Cambium (dried)	**Dom/Hand**: wood used in house building, particularly for roofs; to make containers (used for the preparation of *repùta* and salted meat) and mangers; wood unsuitable to make boards for resting cheese as it releases color and smell; pine needles scattered on the ground to dry the bottom of animal litter; pine needles used as house insulator between interior and exterior walls. **Liq**: immature cones flavouring in liqueur. **Med**: immature cones used to make an antitussive syrup, resin infusion against respiratory system diseases (**resp**) (*); resin (sometimes cooked in oil or butter) to spread on wounds, cracks, and chilblains, due to its antibacterial and healing properties; resin on hematomas, fractures, sprains, and areas affected by gout; dried vascular cambium placed on wounds (**skin**) (musc-skel). **Vet**: resin, heated in butter, to spread on sick hooves, wounds, limb problems and sprains.	**Dom/Hand**: [11,13] **Liq**: [9,10] **Med**: [7,8,9,11,13,16] (**resp**) [5,7,10,13,16] (**skin**) [9] (**musc**) [10] (**skel**) **Vet**: [7,9]	0.54
*Picea abies* (L.) H. Karst. P.abi.HBPNGP_ETN	Abete La peisse/Pèsse (C)	W	Wood (dried) Needles (dried) Young Resinous Buds (fresh or dried) Resin (fresh or dried) Vascular Cambium (dried)	**Al**: hard resin chewed (called *boùilla*) **Dom/Hand**: resin used as glue; resin baked in bone fat to make soap for laundry; wood used in house building, especially interiors, furniture, and to make chests to hold boards for cheese resting, since it does not release color; needle pine scattered on the ground to dry the bottom of animal litter. **Liq**: fresh young resinous buds flavouring in liqueur and grappa. **Med**: fresh young resinous buds used to make an antitussive syrup against respiratory system diseases; resinous bud infusion or decoction against respiratory system diseases and fever (**resp**) (**abn**) (*), and as digestive (**dig**); resin or dried vascular cambium used on wounds (**skin**).	**Al**: [10] **Dom/Hand**: [13] **Med**: [9,10,11,13] (**resp**) [10,11] (**skin**) (**skel**)	0.59
*Pinus cembra* L. P.cem.HBPNGP_ETN	Pino cembro Aròlla	W	Seeds Immature Cones (fresh), Wood (dried)	**Al**: edibles seeds (called *aravés*), ingredient in the *meculìn* recipe. **Dom/Hand**: excellent wood construction, in particular for furniture; easy to work, very fragrant, keeps moths away. **Liq**: immature cones as flavouring in liqueur and grappa. **Med**: immature cones used to make a syrup for respiratory system disease (cough, bronchitis, cold) (**resp**).	**Al**: [5,6,7,10] **Dom/Hand**: [11] **Liq**: [5,6,11] **Med**: [7] (**resp**)	0.65
*Pinus mugo* Turra P.mug.HBPNGP_ETN	Pino montano Pino mugo Pin	W	Young Resinous Buds (fresh), Male Twigs with Pollen Bags Resin Needles	**Liq**: flavouring in liqueur and grappa. **Med**: fresh young resinous buds used to make a syrup for respiratory system diseases (**resp**), resin to spread on wounds and twisting (**skin**) (**musc**). **Others**: dried needles used as tobacco substitute during war.	**Liq**: [6,9,10,11] **Med**: [7,8,9,10,11] (**resp**)	0.35
*Pinus sylvestris* L. P.syl.HBPNGP_ETN	Pino silvestre Daille	W	Young Immature Cones (fresh) Young Resinous Buds (fresh or dried)	**Liq**: immature cones flavouring in liqueur and grappa. **Med**: immature cones used to make a syrup for respiratory system diseases (cough, bronchitis, cold); infusion of resinous buds against respiratory system diseases (**resp**) (*).	**Med**: [7,8,10] (**resp**)	0.13
**Plantaginaceae**						
*Plantago afra* L. syn *P. psyllium* L.	Psillio Gran de natùs (C)	P	Seeds (fresh or dried)	**Med**: used as a remedy for internal hematomas, to dilute the blood: seed ingestion, or seed maceration overnight in water to drink, or wrap with boiled seeds on the hematoma (**circ**) (**musc**); In case of eye problems and inflammations put a seed inside the eye to release mucilage (**eye**).		0.32
*Plantago lanceolata* L. P.lan.HBPNGP_ETN	Piantaggine Plantèn/Piantèn/Pienten (C-V) Foglie plate (V)	W	Leaves (fresh) Floral Stems (fresh)	**Al**: ingredient in soups and omelet (boiled). **Med**: leaves placed on wounds, thorns, burns, infected pimples and insect bites (**skin**); on traumas and hematomas (**musc**); as compress on tooth abscesses (**dent**); decoction or infusion (to drink) against internal inflammation and abdominal pain (**dig**); against cystitis and urinary tract problems (**uri**); disinfectant (**cip**) and anti-inflammatory (**gen**); infusion as cough remedy (**resp**). **Others**: floral stems used to build small chairs as toys for children. **Vet**: leaf rubbed on horsefly stings.	**Al**: [6,7,9,10] **Med**: [5,6,7,8,9,10,11,13](**skin**) [7,10,13] (**musc**) [7] (**uri**) [9,10,11,12] (**resp**)	0.16
*Plantago major* L. P.maj.HBPNGP_ETN		W	Leaves (fresh) Floral Stems (fresh)	As above (see *P. lanceolata*).		0.29
*Plantago media* L. P.med.HBPNGP_ETN		W	Leaves (fresh) Floral Stems (fresh)	As above (see *P. lanceolata*).		0.37
*Veronica fruticans* Jacq. V.fru.HBPNGP_ETN	Tè di montagna	W	Aerial Parts (fresh or dried)	**Med**: thirst-quenching infusion, substitute for tea (**nerv**).		0.03
**Poaceae**						
*Arrhenatherum elatius* (L.) P.Beauv. ex J.Presl and C.Presl. A.ela.HBPNGP_ETN	Retèine Ritta	W	Leaves and Flowers (fresh)	**Vet**: used to make a bed to reinforce weak calves.		0.02
*Avena sativa* L.	Avena	C	Seeds (fresh or dried)	**Al**: seeds ground as flour. **For/Vet**: excellent forage; tonic for horses and donkeys.	**Al**: [13] **For/Vet**: [5,11,13]	0.10
*Cynodon dactylon* (L.) Pers. C.dac.HBPNGP_ETN	Gramigna Lou gramòn	W	Leaves (fresh) Roots (fresh or dried)	**Dom/Hand**: filter made of twisted roots (called *gramòn*) used to remove impurities from milk; also used as a sponge to clean animals. **Med**: radical decoction against prostatitis and urinary tract problems, diuretic (**gen-uri**). **Others**: leaf held between the fingers used as a whistle.	**Med**: [5,7,8,10,13] (**uri**)	0.09
*Festuca ovina* L. F.ovi.HBPNGP_ETN	Eulenna (C)	W	Leaves (fresh or dried)	**For**: used as fodder.		0.02
*Hordeum vulgare* L.	Orzo Orzu (C)	C	Seeds (fresh or dried)	**Al**: seeds ground as flour, ingredient in the barley polenta (called *peilò d’orzo*); roasted and grounded seeds as coffee substitute. **Liq**: seeds maceration in the Cogne beer recipe (with *Berberis vulgaris* and *Polypodium vulgare*) **Med**: seed decoction (to drink) against inflammation and infections of the urinary (**uri**) and digestive (**dig**) systems. **For/Vet**: anti-inflammatory forage; seed decoction added to the mash as anti-inflammatory, in case of infections of the urinary tract, and before and after the cows’ birth.	**Al**: [5,10,11,13] **For/Vet**: [10,11,13]	0.31
*Oryza sativa* L.	Riso Ris (C)	P*	Seeds (fresh or dried)	**Al**: in soups and in risottos. **Med**: against diarrea (**dig**).		0.02
*Secale cereale* L.	Segale Bréla (C)	C	Seeds (fresh or dried)	**Al**: seeds grounded as flour, to make bread. **For**: to feed and fatten animals.	**Al**: [5,10,11,13] **For**: [7,10,11]	0.22
*Stipa pennata* L. S.pen.HBPNGP_ETN	Lino delle fate Pianta segnatempo Plante du temps Le menìn (C) Meneun (R)	W	Aerial Parts (fresh or dried) Roots (fresh or dried)	**Dom/Hand**: hanged on houses’ doors as a timepiece plant, curls up with the arrival of bad weather; twisted root filter (called *gramòn*) used to remove impurities from the milk and as a sponge to clean animals.		0.29
*Triticum* sp.	Frumento Frùmen (C)	C	Seeds (fresh or dried)	**Al**: seeds grounded as flour, to make bread. **For/Vet**: to feed and fatten animals and as galactogogue; given to cows before calving.	**Al**: [10,11,13] **For/Vet**: [7,10,13]	0.21
*Zea mays* L.	Mais Granoturco	P*	Seeds (fresh or dried)	**Al**: flour, ingredient in different polenta recipe (*peilà nèira* in water, and *peilà blàntse* in milk). **For/Vet**: to feed and fatten animals and as galactogogue; liquid polenta given to cows after calving.	**Al**: [5,10,11,13] **For/Vet**: [5,10,11]	0.87
**Polygonaceae**						
*Bistorta officinalis* Delarbre syn *Persicaria bistorta* (L.) Samp. B.off.HBPNGP_ETN	Bistorta Biavèttes/Biavètta (C) Lenve/Lenva (V-R)	W	Leaves Stems (fresh) Rhyzomes (fresh or dried)	**Al**: ingredient in soups and omelettes; stems sucked as refreshing snacks; root macerate used as rennet (called *buì*) used to make *bròssa*, ricotta and *seràs* after making fontina. **For**: galactogogue fodder.	**Al**: [5,7,10]	0.87
*Polygonum aviculare* L. P.avi.HBPNGP_ETN	Groupet (C) Trèinetta (R)	W	Leaves (fresh)	**Med**: leaf infusion or decoction as diuretic and disinfectant against prostatitis and urinary tract problems (**gen-uri**). **Vet**: leaf decoction for cows vaginal washes to cows after calving (against infections).		0.10
*Rheum rhabarbarum* L.	Rabarbaro	C	Stems (fresh)	**Al**: stems as ingredient in jams. **Med**: stems jam as laxative (**dig**).	**Al**: [10] **Med**: [10] (**dig**)	0.09
*Rumex acetosa* L. R.ace1.HBPNGP_ETN	Acetosa Enseuille/Euseille (C) Bruschetta Bruscheun (R)	W	Leaves and Stems (fresh)	**Al**: leaves as ingredient in soups and omelettes; stems eaten as refreshing snacks; leaves used to make a rennet (called *bunì*). **For**: hens fodder increasing eggs production.	**Al**: [5,6,7,9,10,11,13,16]	0.50
*Rumex acetosella* L. R.ace2.HBPNGP_ETN	Acetosa Enseuille/Euseille (C) Bruschetta, Bruscheun (R)	W	Leaves (fresh)	**Al**: Leaves eaten as refreshing snacks; leaves used to make a rennet (called *bunì*).		0.37
*Rumex alpinus* L. R.alp.HBPNGP_ETN	Rabarbaro selvatico Lavache (C-R)	W	Leaves and Stems (fresh)	**Al**: stems as an ingredient in jams or as refreshing snacks. **Dom/Hand**: big leaves used to wrap the shapes of butter and carry them down from mountain pastures to villages. **Med**: stem decoction as laxative (**dig**). **Vet**: leaf decoction as emollient before cows calving.	**Al**: [5,6,7,10,11] **Med**: [6] (**dig**)	0.32
**Polypodiaceae**						
*Dryopteris filix-mas* (L.) Schott D.fil.HBPNGP_ETN	Felce maschio Earba di serpen	W	Leaves (fresh)	**Med**: leaf ingestion as vermifuge (also against tapeworm) (**cip**).		0.02
*Polypodium vulgare* L. P.vul.HBPNGP_ETN	Felce liquirizia Refiòuze/Redefioùzo (C) Rei due (V-R)	W	Roots (fresh or dried)	**Al**: sweet-tasting roots as snack, with licorice flavor. **Liq**: ingredient in the Cogne beer recipe (with *Hordeum vulgare* and *Berberis vulgaris*). **Med**: roots eaten raw as appetizer and against nutritional diseases (**dig**) (**enm**).	**Al**: [7,9,10,11,13,16] **Med**: [7,9] (**dig**)	0.50
**Primulaceae**						
*Primula pedemontana* E.Thomas ex Gaudin P.ped.HBPNGP_ETN	Fieur di mon			Plant whose dialectal name is known.		0.02
*Primula veris* L. P.ver.HBPNGP_ETN	Primula Campanin tsàno (C) Fieur de paque (C) Paquerette (C-V-R)	W	Leaves (fresh) Flowers (fresh or dried)	**Al**: leaves and flowers as ingredients in salads and soups. **Med**: relaxing flower infusion against nervousness (**nerv**).	**Al**: [6,7,9,10,11]	0.31
**Ranunculaceae**						
*Clematis vitalba* L. C.vit.HBPNGP_ETN	Vòble (R)	W	Branches (fresh or dried)	**Dom/Hand**: branches used to build baskets and *gerle*.		0.03
*Pulsatilla alpina* (L.) Delarbre P.alp.HBPNGP_ETN	Fieur de fer	W		Plant whose dialectal name is known.		0.02
*Ranunculus kuepferi* Greuter and Burdet R.kue.HBPNGP_ETN	Erba du boùnèi	W	Aerial Parts (fresh)	**Al**: plant macerated to make a rennet (called *boùnèi*).		0.02
*Ranunculus montanus* Willd. R.mon.HBPNGP_ETN	Ranuncolo montano	W	Aerial Parts (fresh)	**Al**: to curdle milk.		0.06
*Trollius europaeus* L. T.eur.HBPNGP_ETN	Bouton d’or	W		Plant whose dialectal name is known.		0,02
**Rosaceae**						
*Alchemilla xanthochlora* Rothm. syn *A. vulgaris* L. A.vul.HBPNGP_ETN	Alchemilla Goubelette (C) Porta rusò (C)	W	Leaves (fresh) Exudate Water Stems (fresh)	**For**: galactogogue fodder. **Med**: walking on leaves against nervousness, fatigue, and depression (**nerv**); refreshing and soothing against foot pain and swelling (**skin**). **Vet**: stems placed on cow’s infected breast.	**For**: [9,16] **Med**: [9] (**nerv**)	0.09
*Amelanchier ovalis* Medik. A.ova.HBPNGP_ETN	Pero corvino Metzéròn/Meutseuron (C)	W	Fruits (fresh or dried)	**Al**: edibles fruits.	[7,13]	0.29
*Aria edulis* (Willd.) M.Roem. syn *Sorbus aria* (L.) Crantz A.edu.HBPNGP_ETN	Sorbo domestico Tumé (C) Ènsàlle (R) Ansalit (V)	W	Fruits (dried	**Al**: dried fruits ground as flour, used to make bread in time of war.	[5]	0.09
*Aruncus dioicus* (Walter) Fernald A.dio.HBPNGP_ETN	Barba di capra Asparago di montagna	W	Young Shoots (fresh)	**Al**: young shoots stir-fried with butter, added with eggs.	[7,10,13]	0.04
*Filipendula ulmaria* (L.) Maxim. F.ulm.HBPNGP_ETN	Reine des prés	W	Leaves and Flowers (fresh or dried)	**Med**: leaf and flower infusions as disinfectant, diuretic, and purifying, against fever, prostatitis, urinary infections, cystitis, kidney stones, joint pain, and rheumatism (**dig**) (**gen-uri**) (**musc**) (**circ**) (**abn**). **Vet**: leaves as forage against joint problems.	[11]	0.18
*Fragaria vesca* L. F.ves.HBPNGP_ETN	Fragolina di bosco Emfré, Pequin (C) Frèie (R), Enfrèies (V)	W	Fruits (fresh) Leaves (fresh or dried)	**Al**: fruits eaten raw or as ingredient in jams. **Med**: leaf infusion as diuretic (**uri**); leaves on wounds as hemostatic (**skin**) (**circ**).	**Al**: [5,6,7,9,10,11,13]	0.69
*Malus domestica* (Suckow) Borkh.	Melo Pomme	P	Fruits (fresh) Seeds (dried)	**Al**: edible fruits. **Liq**: seeds flavouring in liqueur.	**Al**: [10,13]	0.03
*Prunus avium* L.	Ciliegio selvatico Cériesei (C)	C	Fruits (fresh) Peduncles (dried)	**Al**: edible fruits. **Med**: infusion of fruit peduncles as diuretic (**uri**).	**Al**: [5,6,7,10,13] **Med**: [7,10,13] (**uri**)	0.04
*Prunus amygdalus* Batsch syn *Prunus dulcis* (Mill.) D.A.Webb	Mandorlo Mèndulei (C)	P	Fruits (fresh)	**Al**: edible fruits. **Med**: fruit decoction against bellyache (**dig**) and menstrual pain (**gen**).		0.02
*Rosa* × *alba* L.	Rosa bianca	C	Flowers (fresh or dried)	**Med**: flower infusion (as compress or wash) on inflamed eyes (**eye**).		0.12
*Rosa canina* L. R.can.HBPNGP_ETN	Rosa canina Eulièntse Gratta cù	W	Fruits Flowers (fresh or dried)	**Al**: fruits eaten raw or as an ingredient in jams and syrups. **For**: fruits given to goats, calves, and pigs. **Liq**: fruits flavouring in grappa. **Med**: fruit infusion (to drink) as vitaminizing and invigorating (**enm**); fruit infusion as antidiarrheal and against digestive problems (**dig**), against respiratory diseases (**resp**) (**abn**); fruit decoction against rheumatic pains (**musc**) and circulatory diseases (**circ**); compress of flower infusion on inflamed eyes and skin (**eye**) (**skin**).	**Al**: [5,6,7,9,11,13] **Liq**: [9,10,13] **Med**: [5,7,9,11] (**dig**) [7,8] (**eye**)	0.77
*Rosa* × *centifolia* L.		C	Flowers (fresh or dried)	**Med**: flower syrup against respiratory diseases (**resp**).	[7]	0.02
*Rubus idaeus* L. R.ida.HBPNGP_ETN	Lampone Pequin (C) Empouè (C)	W	Fruits (fresh)	**Al**: fruits eaten raw or as an ingredient in jams. **Liq**: fruits flavouring in grappa.	**Al**: [5,6,7,9,10,11,13] **Liq**: [6,10]	0.81
*Rubus saxatilis* L. R.sax.HBPNGP_ETN	Queglierettes (C-V) Grosàlles (R)	W	Fruits (fresh)	**Al**: fruits eaten raw or as an ingredient in jams.		0.32
*Sorbus aucuparia* L. S.auc.HBPNGP_ETN	Sorbo degli uccellatori Toumé (C)	W	Fruits (fresh)	**For**: fruits as feed for blackbirds	[5,13]	0.02
**Rubiaceae**						
*Coffea* sp.	Caffè	P*	Seeds (dried)	**Med**: a teaspoon of coffee mixed with sugar (to ingest) or infusion (to drink) against headache and mestrual pains diseases (**nerv**) (**enm**) (**gen**). **Vet**: given to cows to stimulate rumen in case of digestive problems.		0.06
*Galium verum* L. G.ver.HBPNGP_ETN	Caglio Erba du caille Earba du cail (C)	W	Flowers (fresh or dried) Leaves (fresh)	**Al**: used as rennet to curdle milk, put directly in milk, or as decoction or macerate.	[7]	0.21
*Galium lucidum* All. G.luc.HBPNGP_ETN	Caglio Erba du caille	W	Flowers (fresh or dried) Leaves (fresh)	As below (see *G. verum*).		0.02
*Galium mollugo* L. G.mol.HBPNGP_ETN	Caglio Erba du caille	W	Flowers (fresh or dried) Leaves (fresh)	As below (see *G. verum*).		0.02
*Rubia tinctorum* L.	Robbia	P	Roots (fresh or dried)	Dom/Hand: root decoction used to dye wool (yellow-red).		0.02
**Salicaceae**						
*Salix babylonica* L.	Salice piangente	P	Bark (fresh or dried)	**Med**: bark infusion or decoction (to drink) against fever (**abn**). **Others**: bark (detached from branches after soaking in water) used to build whistles.		0.02
*Salix caprea* L. S.cap.HBPNGP_ETN	Salice Sàrdzu (C) Gouras (C)	W	Young Branches (fresh or dried)	**Dom/Hand**: young branches used to build baskets and *gerle*.		0.04
*Salix purpurea* L. S.pur.HBPNGP_ETN	Salice Sàrdzu (C)	W	Young Branches (fresh or dried)	As above (see *S. caprea*).		0.09
**Sapindaceae**						
*Acer pseudoplatanus* L. A.pse.HBPNGP_ETN	Plòno Plòne	W	Bark and Wood (dried)	**Dom/Hand**: wood used to build cheese shapes and cheese boards. **Vet**: bark decoction as antidiarrheal for calves.		0.06
**Scrophulariaceae**						
*Scrophularia nodosa* L. S.nod.HBPNGP_ETN	not reported	W	Leaves (fresh)	**Med**: fresh leaves placed on wounds (**skin**)		0.02
*Verbascum thapsus* L. V.tha.HBPNGP_ETN	Tasso barbasso	W	Roots (fresh or dried)	**Med**: root decoction (to drink) as depurative against liver problems and bellyache (**dig**).		0.02
**Solanaceae**						
*Solanum lycopersicum* L.	Pomodoro Tumattes (C)	C/P	Fruits (immature, fresh)	**Al**: immature fruits as ingredient in jams.		0.06
*Solanum tuberosum* L.	Patata Tertéuffie (C)	C	Tubers (fresh) Leaves (dried)	**Al**: ingredient in soups, boiled, or stir-fried with butter. **For**: tubers and leaves added to the mash, as galactogogue and to make milk fatter. **Med**: raw potato slice placed on irritated eyes, burns or hemorrhoids (**eye**) (**skin**) (**circ**). **Others**: dried leaves used as tobacco substitute.		0.21
**Thymelaceae**						
*Daphne mezereum* L. D.mez.HBPNGP_ETN	Dafne Pequin di serpen (C)	W	Fruits (fresh or dried)	**Med**: poisonous fruits ingestion to cause abortion (**pcp**).		0.02
**Tropaeolaceae**						
*Tropaeolum majus* L.	Nasturzio	C	Seeds (dried)	**Med**: seed infusion as antidiarrheal (**dig**).		0,03
**Urticaceae**						
*Parietaria officinalis* L. P.off.HBPNGP_ETN	Parietaria Marquiroù (C)	W	Leaves (fresh or dried)	**Med**: leaves placed on nettle stings to alleviate itch (**skin**).		0.02
*Urtica dioica* L. U.dio.HBPNGP_ETN	Ortica Ortchìe Ortché (C)	W	Roots (fresh or dried) Leaves (fresh or dried)	**Al**: ingredient in soups, omeletes, in the dough of the dumplings; macerate used as rennet to curdle milk. **Cosm**: root or leaf decoction (as compresses or wash) to fortify, polish, degrease hairs and as anti-dandruff. **Dom/Hand**: placed against moths. **For**: dried leaves given to hens for increasing eggs production. **Med**: leaf infusion or decoction as depurative and tonic (**uri**)(**dig**), galactogogue (**pcp**), against anemia; pricking with nettle to reactivate circulation (**circ**). **Others**: macerate used as fertilizer and pesticide (spread on plants). **Vet**: dried leaves given to cows in case of digestive problems.	**Al**: [5,6,7,9,10,11,13] **Cosm**: [6,8,9,10,11,13,16] **For**: [7,11,13,16] **Med**: [5,6,7,9,10] (**uri**) [6,9,10,13] (**circ**) **Others**: [13]	0.91
**Verbenaceae**						
*Aloysia citrodora* Palau	Erba luisa Erba limonina	P	Leaves (dried)	**Med**: digestive infusion (**dig**) (non-ancient use).		0.02
*Verbena officinalis* L.	Verbena	P	Leaves (fresh or dried)	**Med**: relaxing infusion, against restlessness and insomnia (**nerv**).		0.02
**Viburnaceae**						
*Sambucus nigra* L. S.nig.HBPNGP_ETN	Sambuco Sayeus Sahù (C)	W	Flowers (fresh or dried) Fruits (fresh) Branches (fresh)	**Al**: flowers’ refreshing drink; fresh flowers consumed in batter; fresh fruits consumed raw or processed in jams or juices. **Dom/Hand**: branches used to build handles of baskets and traditional *gerle*. **Liq**: elderberry wine: flowers maceration in water and lemon. **Med**: fruit jam or syrup, or floral infusion, against fever and cough (**resp**) (**abn**); fruit jam or syrup against digestive problems (**dig**); floral infusion compress on irritated eyes (**eye**). **Vet**: fresh flowers given to cows against digestive problems.	**Al**: [5,6,7,9,10,11,13] **Med**: [8,9,10,11,13] (**resp**) [10,16] (**dig**) [6,13] (**eye**)	0.24
*Sambucus racemosa* L. S.rac.HBPNGP_ETN	Savu (C)	W		Plant whose dialectal name is known.		0,02
*Viburnum lantana* L. V.lan.HBPNGP_ETN	Lentàna (R) Lantana (V)	W	Branches (fresh)	**Dom/Hand**: branches used to build handles of baskets and traditional *gerle*.		0.09
**Violaceae**						
*Viola calcarata* L. V.cal.HBPNGP_ETN	Viola di montagna Viola alpina Vieulette/vieuletta (C)	W		**For**: galactogogue fodder, making fontina particularly fat and valuable. **Liq**: flowers flavouring in grappa. **Med**: infusion (to drink or to inhale) or syrup against flu, cough, catarrh, sore throat, bronchitis, cold and respiratory problems (**resp**) (**abn**). (*) Attention high doses can cause nose bleeding; recommended dose 4–5 flowers per cup.	**Med**: [7]	0.75
*Viola odorata* L. V.odo.HBPNGP_ETN	Viola	W	Flowers (fresh or dried)	**Al**: flowers added in salads; candied.	[6,13]	0.03
*Viola tricolor* L. V.tri.HBPNGP_ETN	Viola	W	Flowers (fresh or dried)	**Med**: floral infusion against cough (**resp**).	[6,7,10] (**resp**)	0.02
Vitaceae						
*Vitis vinifera* L.	Vite Rèsìn (C)	P	Fruits (fresh)	**Liq**: ingredient in *vin brulè*, a wine boiled and flavuored with cloves and cinnamon; raisins ingredients in *meculìn* recipe. **Med**: hot *vin brulé* (to drink) against fever and colds (**resp**) (**abn**).	**Liq**: [13] **Med**: [13]	0.03
**NON-PLANT-BASED FOODS AND REMEDIES**						
**LICHENES**						
**Parmeliaceae**						
*Cetraria islandica* (L.) Ach.	Lichene Lichene islandico	W	Thallus (dried)	**Med**: decoction (to drink or as compress) or syrup against respiratory disease, phlegm, bronchitis and cough (**resp**). (*) Attention high doses can cause health problems, it is recommended to eliminate the first decoction water and repeat the procedure.	[5,6,7,9,10,11,12] (**resp**)	
**FUNGI**						
**Agaricaceae**						
*Calvatia gigantea* Lloyd	Vescia maggiore Boulé blanc	W	Basidiocarp (fresh)	**Al**: sliced fried in batter.	[10]	
*Lycoperdon perlatum* Pers.	Vescia	W	Basidiocarp (fresh) Spores (dried)	**Al**: consumed fresh or preserved in oil or vinegar. **Med**: Spores to spread on wounds, burns, and warts, as healing and disinfectant (**skin**).		
*Macrolepiota procera* Singer	Mazza di Tamburo	W	Basidiocarp (fresh)	**Al**: consumed cooked.	[10]	
**Boletaceae**						
*Boletus edulis* Bull.	Porcino	W	Basidiocarp (fresh or dried)	**Al**: consumed fresh or preserved in oil or vinegar, or dried	[5,10]	
*Leccinum scabrum* (Bull.) Gray	Porcinella	W	Basidiocarp (fresh or dried)	**Al**: consumed fresh or preserved in oil or vinegar, or dried		
**Cantharellaceae**						
*Cantharellus cibarius* Fr.	Galletti Margherite	W	Basidiocarp (fresh)	**Al**: ingredient in risotto or stir-fried with other mushrooms.	[10]	
**Clavariaceae**						
*Ramaria botrytis* (Pers.) Bourdot	Manine rosa	W	Basidiocarp (fresh)	**Al**: consumed fresh or preserved in oil or vinegar, or dried		
**Fomitopsidaceae**						
*Fomitopsis officinalis* (Vill.) Bondartsev and Singer	Fungo del larice Boulì des plantes	W	Basidiocarp (fresh or dried)	**Liq**: flavoring in liqueur for its bitter properties.		
**Suillaceae**						
*Suillus granulatus* (L.) Roussel	Prataiolo di bosco	W	Basidiocarp (fresh)	**Al**: consumed fresh, stir-fried with other mushrooms, or preserved in oil or vinegar.		
*Suillus grevillei*(Klotzsch) Singer	Prataiolo	W	Basidiocarp (fresh)	**Al**: consumed fresh, stir-fried with other mushrooms, or preserved in oil or vinegar.	[10]	
**OTHERS**						
Insect galls	Coutonettes			**Med**: galls on *Rosa canina* and *Berberis vulgaris* placed and rubbed on wounds as haemostatic (**skin**).		
Ibex marrow				**Med**: spread on hematomas (**musc**); smeared on the chest or ingested to remove phlegm (**resp**).		
Marmot fat				**Med**: spread on hematomas (**musc**); to treat skeletal problems (**skel**); smeared on the chest or ingested to remove phlegm (**resp**).		
Quinine	Chinìno			**Med**: used in the past against fever.		
Theriaca	Triàca, Triàcca			**Med**: used in the past as *panacea* for several diseases.		
Honey				**Med**: spread on hematomas (**musc**); to treat skin diseases (e.g., bruises and plugs) (**skin**).		
Snake skin				**Med**: to treat skin diseases (e.g., plugs) (**skin**).		
Breast milk				**Med**: to treat ears diseases (**ear**).		
Mud	Paciòqque			**Med**: to treat skin problems (e.g., gadfly bites) (**skin**)		
raw wool				**Med**: bedtime in case of high fever or pneumonia (**resp**); against arthritis and musculoskeletal pain (**musc**).		

^a^ Dialectal names used in each valley (C): Cogne (V): Valsavarenche (R): Rhêmes. ^b^ W, wild; C, cultivated; P, purchased from adjacent areas at lower altitude (from Aosta Valley (e.g., Aymavilles, Introd or from Piedmont)). P*, commercially purchased from distant areas. ^c^ In the table ‘Flowers’ are also intended as inflorescences; ‘Roots’ are also intended as Rhizomes. ^d^
**Use Categories**: **Al**: Alimentary; **Cosm**: Cosmetic; **Dom/hand**: Domestic and handicraft; **For**: Forage; **Liq**: Liquoristic; **Med**: Medicinal; **Others**: Others; **Vet**: Veterinary. **Medicinal subcategories**: (**abn**) abnormal symptoms, signs not elsewhere classified (including fever); (**cip**) certain infections and parasitosis; (**circ**) circulatory system; (**dent**) dental and oral; (**ear**) ear and mastoid process; (**eye**) eye and adnexa; (**dig**) digestive tract; (**enm**) endocrine, nutritional and metabolic; (**gen-uri**) genitourinary system; (**musc-skel**) musculoskeletal system and connective tissue; (**nerv**) nervous system; (**pcp**) pregnancy, childbirth and puerperium; (**resp**) respiratory tract; (**skin**) skin and subcutaneous tissues. ^e^ Bibliographic comparison with several ethnobotanical researches carried out on Western Italian Alps: Binel, 1972 [15]; Signorini and Fumagalli [16], 1983; Pieroni et al., 2009 [5]; Vitalini et al., 2009 [8]; Mattalia et al., 2012 [6]; Vitalini et al., 2013 [9]; Cornara et al., 2014 [13]; Vitalini et al., 2015 [10]; Bellia and Pieroni, 2015 [7]; Dei Cas et al., 2015 [11]; Bottoni et al., 2020 [12]. ^f^ Local importance of each species on the basis of the Relative Frequency of Citation. (*) Typical mix of plants to treat respiratory diseases: *Salvia officinalis*, *Viola calcarata* and *Cetraria islandica*. Sometimes added: *Tussilago farfara*, *Artemisia genipi*, Pinaceae needles, resin or resinous buds.

**Table 3 plants-11-00170-t003:** Species with a Relative frequency of Citation (RFC) > 0.50.

Species	FC	NC	RFC	Species	FC	NC	RFC
*Peucedanum ostruthium*	66	4	0.97	*Artemisia genipi*	50	4	0.74
*Urtica dioica*	62	7	0.91	*Artemisia absinthium*	49	5	0.72
*Blitum bonus-henricus*	61	1	0.90	*Achillea erba-rotta*	48	3	0.71
*Juniperus communis*	60	5	0.88	*Fragaria vesca*	47	2	0.69
*Bistorta officinalis*	59	2	0.87	*Linum usitatissimum*	46	2	0.68
*Vaccinium myrtillus*	58	3	0.85	*Pinus cembra*	44	4	0.65
*Taraxacum officinale aggr*.	55	4	0.81	*Tussilago farfara*	41	1	0.60
*Rubus idaeus*	55	2	0.81	*Picea abies*	40	4	0.59
*Malva neglecta*	55	3	0.81	*Gentiana punctata*	38	5	0.56
*Arnica montana*	53	3	0.78	*Carum carvi*	38	3	0.56
*Rosa canina*	52	4	0.77	*Larix decidua*	37	4	0.54
*Berberis vulgaris*	52	5	0.77	*Rumex acetosa*	34	2	0.50
*Viola calcarata*	51	3	0.75	*Polypodium vulgare*	34	3	0.50

FC, number of informants mentioning the species; NC, number of categories of use; RFC, relative frequency of citation.

**Table 4 plants-11-00170-t004:** Factor informant consensus (Fic) index related to the agreement on species used to treat different diseases.

Disease Subcategories	Number of Citations	Number of Species	Fic
Respiratory tract	352	36	0.90
Musculoskeletal system and connective tissue	156	25	0.85
Skin and subcutaneous tissues	219	36	0.84
Digestive system	315	52	0.84
Genitourinary tract	169	32	0.82
Sensory system	62	13	0.80
Dental and oral	33	9	0.75
Circulatory system	79	23	0.72
Pregnancy, childbirth and the puerperium	50	15	0.71
Infections and parasitosis	50	16	0.69
Nervous system	73	23	0.69
Symptoms and signs not elsewhere classified	40	19	0.54
Endocrine, nutritional and metabolic	31	15	0.53

**Table 5 plants-11-00170-t005:** Fidelity level (FL) value of medicinal plants against a given disease subcategory and main bioactive compounds responsible for the therapeutic effects.

Species	FL	NI	NC	Disease Subcategories	Main Bioactive Compounds
*Cetraria islandica*	100%	28	30	Respiratory tract	Lichen polysaccharides with antiviral [28] and anti-inflammatory properties [29]
*Pinus cembra*	100%	19	19	Respiratory tract	Phenolic acids, flavonoids, proanthocyanidins with antimicrobial activities [30]
*Arctostaphylos uva-ursi*	100%	18	18	Genitourinary tract	Hydroquinones and tannins with antiseptic and antimicrobial properties [31]
*Chelidonium majus*	100%	10	11	Skin and subcutaneous tissues	Isoquinoline alkaloids, flavonoids and phenolic acids with antibacterial, antiviral, and antifungal properties [32]
*Tanacetum vulgare*	100%	10	11	Infections and parasitosis	β-Thujone with anthelmintic activity [33]
*Pinus sylvestris*	100%	9	10	Respiratory trat	Terpenoids, steroids, proanthocyanidins, flavonoids and resin acids with anti-inflammatory, anti-bacterial, antiseptic properties [34,35]
*Viola calcarata*	98.30%	49	58	Respiratory tract	Flavonoids, mucilage [36]
*Pinus mugo*	95.80%	24	24	Respiratory tract	δ-3-Carene; α-pinene; (E)-caryophyllene; limonene; β-pinene, linalool acetate, germacrene and linalool with anti-inflammatory [37] and antimicrobial activities [35]
*Tussilago farfara*	91.70%	41	48	Respiratory tract	Sesquiterpenes, triterpenoids, flavonoids, phenolic acids, chromones and its derivatives, alkaloids with anti-inflammatory and anti-microbial activities [38]
*Allium sativum*	85.70%	20	21	Infections and parasitosis	Sulfur-containing phytoconstituents and flavonoids with antibacterial, antiviral, antifungal, antiprotozoal, antioxidant and anti-inflammatory activities [39]
*Plantago major*	84.60%	13	13	Skin and subcutaneous tissues	Polysaccharides responsible of wound healing effects [40]
*Gentiana punctata*	82.80%	22	29	Digestive system	Polyphenol such as flavones and their glycosides with gastroprotective activity [41]
*Plantago media*	78.40%	24	37	Skin and subcutaneous tissues	Polysaccharides and flavonoids with wound healing and anti-inflammatory effects [42]
*Arnica montana*	76.40%	53	72	Musculoskeletal system and connective tissue	Sesquiterpene lactones of helenalin and dihidrohelenalin type, essential oil, flavonoids, phenolic acids [43] with anti-inflammatory activity [44,45]
*Juniperus communis*	72.50%	50	80	Digestive system	Flavonoids, essential oil and coumarins with hepatoprotective, anti-inflammatory and antimicrobial activities [46], as well as digestive properties [47]

## Data Availability

The data presented in this study are available in this article; databases are held at Distav (University of Genoa) and at PNGP (Aosta).

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
