# Peer review of "Ethnomedicinal and Ethnobotanical Survey in the Aosta Valley Side of the Gran Paradiso National Park (Western Alps, Italy)"

_plants, 2022, doi:10.3390/plants11020170_

Round 1

Reviewer 1 Report

The authors of the review titled "Ethnobotanical and ethnomedicinal uses of plants in the Gran Paradiso National Park, Aosta Valley (Western Alps, Italy)" did a extensive compilation of the use of different medicinal plants and other organisms  in a particular region with considerable used of natural resources with pharmacological value. The manuscript is well written and the tables are well presented, Personally I would like to find more about the chemicals presents in the plants, for example another column in the table 2. But it is a suggestion, because there are massive information in the review.

Other than that suggestion the manuscript is great

Author Response

We thank the reviewer for his useful comment, however some additional information on the main bioactive compounds of plants that obtained the highest FL had been included in Table 5. Therefore, to avoid stretching the work too much we have not included an additional column in table 2. On the other hand, another reviewer pointed out that the manuscrict is already too extensive, so we did not consider expanding it further. Further studies are in program to highlight phytochemical and biological properties of some selected species used in the traditional folk medicine of the PNGP.

Reviewer 2 Report

I think this article would be better suited for a 
Special Issue "Linking Biodiversity and Cultural Diversity: New Approaches for Fostering the Sustainable Use of Natural Resources" Biology MDPI.

I enclose the results of the plagscen

Author Response

Thank you for your suggestion but regarding this we specify that in case of acceptance of the paper we will take advantage of a voucher for publication on the Topic collection of Plants MDPI: “Bioactive Compounds in plants”.

Reviewer 3 Report

Plants (Manuscript ID: plants-1524754), Comments to the Authors:

Title: Ethnobotanical and ethnomedicinal uses of plants in the Gran 2 Paradiso National Park, Aosta Valley (Western Alps, Italy)

Comments

The submitted manuscript summarized dialogues and semi-structured interviews with 68 native informants (30 men, 38 women; mean age 70) carried out between 2017 and 2019. A total of 39 reports have been collected, concerning 217 taxa (including 10 mushrooms, 1 lichen) mainly used for medicinal (42%) and food (33%) purposes. Minor uses were related to liquor making (7%), domestic (7%), veterinary  (5%), forage (4%), cosmetic (1%) and other (2%). Medicinal plants were used to treat 14 ailment categories, of which the most important were respiratory (22%), digestive (19%), skin (13%), musculoskeletal (10%) and genitourinary (10%) diseases. Data were also evaluated by quantitative ethnobotanical indexes. The results show a rich and alive traditional knowledge concerning plants uses in the Gran Paradiso National Park. Plants resources may provide new opportunities from the scientific point of view, for the valorization of local products for health community and for sustainable land management.

I think the submitted manuscript can be accepted after the authors respond to the following comments:

  1. What is the difference between ethnobotanical and ethnomedicinal in the context of the submitted manuscript.
  2. The authors investigate mushrooms that are not plant so the title should be changed.
  3. The authors should compare their work to previous work on nearby locations and provide useful correlations.
  4. The authors should include a future perspective section to indicate the application of their work.  

Author Response

  1. What is the difference between ethnobotanical and ethnomedicinal in the context of the submitted manuscript.

We have added a phrase in the Introduction to better explain the difference between ethnobotanical and ethnomedicinal studies carried out in this contest:

“Our survey combined ethnobotanical data concerning the relationships between people and plants with folk medicine, to preserve the traditional uses of the local flora and to revitalize the strong cultural identity of the alpine valleys. The recovery of TEK has an intrinsic cultural value and mainly provide some new opportunities for a sustainable land management and for the valorization of local products [24]. In addition, data obtained on traditional use of natural products could be useful for the rational development of new medicines for health community.”

  1. The authors investigate mushrooms that are not plant so the title should be changed.

Following the suggestion of the reviewer we haven’t change the Title but we have moved all the data concerning fungi, lichens and non-plant based remedies to the bottom of Table 2 under the heading "NON-PLANT-BASED FOODS AND REMEDIES".

However, In our study the number of mushrooms cited is very low and in addition the insertion of some taxa belonging to fungi and lichens is a common practice in most ethnobotanical surveys (see for example Vitalini et al 2015 J. of Ethnopharm. 173:435-458 “Plants, people and traditions: Ethnobotanical survey in the Lombard Stelvio National Park…)”; La Rosa et al 2021 J. of Ethnobiol and Ethnomed 17:47 “Ethnobotany of the Aegadian Islands…”).

  1. The authors should compare their work to previous work on nearby locations and provide useful correlations.

A bibliographic comparison with similar uses in North-Western Italian Alps is provided in a column in Table 2. The comparison with the following papers has been made: Binel, 1972; Signorini and Fumagalli, 1983; Pieroni et al., 2009; Vitalini et al., 2009; Mattalia et al., 2012; Vitalini et al., 2013; Cornara et al., 2014; Vitalini et al., 2015; Bellia and Pieroni, 2015; Dei Cas et al., 2015, Bottoni et al., 2020. Some correlations with traditional uses of plants in the neighbouring areas are showed in the Discussion.

  1. The authors should include a future perspective section to indicate the application of their work.  

We have added, in the Conclusions, some considerations about the future perspective of our work:

“As a future perspective, it would be useful to complete the ethnobotanical investigation also in the Piedmont side of the PNGP. It would also be useful to select some medicinal species and evaluate the possibility of local production of phytotherapeutics based on traditional remedies. The Park would play a key role in optimizing the ecosystem services of the flora, in a context of greater involvement of local populations.”

We thank the reviewer for his useful comments.

Reviewer 4 Report

Article is well done and it is worth to be published after minor revision. Some comments are give below.

Title

Think to omit Aosta Valley (National Park is enough)

Introduction

Line 39: change hyphen before 'including' into comma

Line 40: change hyphen after 'Park' into comma

Results

Table 2

Blitum bonus-henricus (L.) Rchb: add point after 'Rchb' ('Rchb.' is correct)

Imperatoria ostruthium L.: Peucedanum ostruthium (L.) W.D.J.Koch is a valid name today while Imperatoria ostruthium L. is a synonim. So, please, put 'Peucedanum ostruthium (L.) W.D.J. Koch.' as the main name and 'Imperatoria ostruthium L.' as a synonym

Arctium lappa L.: change 'Grandiflorum' into 'grandiflorum'

Arctium lappa L.: 'Attention not to be confused with the similar species Doronicum grandiflorum Lam. subsp. Grandiflorum.' What does it mean? Arctium lappa and Doronicum grandiflorum are not similar species.

Artemisia campestris L. subsp. borealis (Pall.) H.M.Hall and Clem. and Leontopodium nivale (Ten.) È. Huet et A. Huet ex Hand.-Mazz. subsp. alpinum (Cass.) Greuter: add space before 'Hall' or omit space before 'Huet'. The way of writing should be the same for all plant names. Please, check all other names in table 2.

Centarea cyanus L.: omit 'syn Cyanus segetum Hill' (this synonym has not been used for a long time)

Doronicum grandiflorum Lam. subsp. grandiflorum: if there is no ethnobotanical uses why it is mentioned?

Lonicera caerulea L. subsp. caerulea: if there is no ethnobotanical uses why it is mentioned?

Agrostemma githago L.: if there is no ethnobotanical uses why it is mentioned?

Colchicum bulbocodium Ker Gawl.: if there is no ethnobotanical uses why it is mentioned?

Convolvulus arvensis L.: if there is no ethnobotanical uses why it is mentioned?

Hippophae rhamnoides L.: omit 'syn Elaeagnus rhamnoides (L.) A. Nelson' (this synonym has not been used for a long time)

Juncus jacquinii L.: if there is no ethnobotanical uses why it is mentioned?

Salvia rosmarinus Spenn.: please, put 'Rosmarinus officinalis L.' as the main name and 'Salvia rosmarinus Spenn.' as a synonym

Rhinanthus alectorolophus (Scop.) Pollich: if there is no ethnobotanical uses why it is mentioned?

Abies alba Mill.: if there is no ethnobotanical uses why it is mentioned?

Primula pedemontana E.Thomas ex Gaudin: if there is no ethnobotanical uses why it is mentioned?

Pulsatilla alpina (L.) Delarbre: if there is no ethnobotanical uses why it is mentioned?

Trollius europaeus L.: if there is no ethnobotanical uses why it is mentioned?

Alchemilla vulgaris L.: please, put 'Alchemilla xanthochlora Rothm.' as the main name and 'A. vulgaris L.' as a synonym

Aria edulis (Willd.) M.Roem.: please, omit Aria edulis (Willd.) M.Roem. and keep only Sorbus aria (L.) Crantz. (synonym Aria edulis (Willd.) M.Roem. has not been used for a long time)

Rosa x alba L.: change letter x with symbol ×

Rosa x centifolia L.: change letter x with symbol ×

Sambucus racemosa L.: if there is no ethnobotanical uses why it is mentioned?

Line 117. Add space (row) after text below Figure 1.

Line 133. Add space (row) before Table 3.

Table 3

Change 'Imperatoria ostruthium' with valid name 'Peucedanum ostruthium'

Delete point after 'genipi.'

Line 138: change 'Imperatoria ostruthium' with valid name 'Peucedanum ostruthium'

Lines 154–154: change 'Alchemilla vulgaris' with valid name 'Alchemilla xanthochlora'

Line 213: change 'J. communis' into 'Juniperus communis' (to be uniform with other plant names in this paragraph)

Line 222: it shoud be: Rosa sp., (add point)

Line 238. Add space (row) after text below Figure 3.

Line 246. Change 'L.' into 'Larix'

Line 246. Change 'P.' into 'Picea'

Line 246. Change 'P. cembra' into 'Pinus cembra'

Line 249: change 'vulgaris.' into 'vulgaris,'

Line 266: change 'T. officinale' into 'Taraxacum officinale'

Line 279: change 'Ribes petraeum' into 'R. petraeum'

Line 288: change 'cereal' into 'cereale'

Line 229: change 'U. dioica' into 'Urtica dioica'

Line 229: change 'A. genipi' into 'Artemisia genipi'

Line 303: change 'B. rapa' into 'Brassica rapa'

Line 303: change 'S. tuberosum' into 'Solanum tuberosum'

Line 304: it shoud be: Triticum sp., (add point)

Line 309: change 'V. calcarata' into 'Viola calcarata'

Line 311: change 'L. usitatissimum' into 'Linum usitatissimum'

Line 314: change 'Cardus' into 'Carduus'

Line 317: change 'G. punctata' into 'Gentiana punctata'

Line 318: What is A. montana? Arnica montana?

Line 321: change 'T. officinale, T. pratense and U. dioica' into ''Tarxacum officinale, Ttrifolium. pratense and Urtica dioica'

Lines 331, 338, 340, 352, 358, 423, 434, 451, 455, 593: as above to avoid confusion amon so many species

Line 387: Rosa x alba L.: change letter x with symbol ×

Lines 438, 442, 594: change 'I. ostruthium' with valid name 'Peucedanum ostruthium'

Author Response

We thank the reviewer for his useful comments.

As written in chapter 2.2 "The nomenclature of plants follows “Plants of the world” and the corresponding synonymous were added, according to “Flora d’Italia” [59,60]. “Index fungorum” [61] was used for the nomenclature of fungal species.”

Some synonyms, even if no longer used for a long time, are reported in order to better find old herbarium specimens in the Ethnobotanical Herbarium of the PNGP.

Title

Think to omit Aosta Valley (National Park is enough)

The PNGP extends over two regions: Piedmont and Aosta Valley. The study was conducted on the Aosta Valley side, which shows a peculiar ethnic identity.  However, now the title has been changed to better explain this point.

Introduction

Line 39: change hyphen before 'including' into comma Done

Line 40: change hyphen after 'Park' into comma Done

 Results

Table 2

Blitum bonus-henricus (L.) Rchb: add point after 'Rchb' ('Rchb.' is correct) Done

 Imperatoria ostruthium L.: Peucedanum ostruthium (L.) W.D.J.Koch is a valid name today while Imperatoria ostruthium L. is a synonim. So, please, put 'Peucedanum ostruthium (L.) W.D.J. Koch.' as the main name and 'Imperatoria ostruthium L.' as a synonym Done

 Arctium lappa L.: change 'Grandiflorum' into 'grandiflorum' Done

Arctium lappa L.: 'Attention not to be confused with the similar species Doronicum grandiflorum Lam. subsp. Grandiflorum.' What does it mean? Arctium lappa and Doronicum grandiflorum are not similar species. Done: the note has been deleted

 Artemisia campestris L. subsp. borealis (Pall.) H.M.Hall and Clem. and Leontopodium nivale (Ten.) È. Huet et A. Huet ex Hand-Mazz. subsp. alpinum (Cass.) Greuter: add space before 'Hall' or omit space before 'Huet'. The way of writing should be the same for all plant names. Please, check all other names in table 2. Done: we omitted spaces, according to “Plants of the World”. We have also standardized “et, and, &” using only &

Centaurea cyanus L.omit 'syn Cyanus segetum Hill' (this synonym has not been used for a long time) Done: The synonymous has been omitted

Doronicum grandiflorum Lam. subsp. grandiflorum: if there is no ethnobotanical uses why it is mentioned?

Concerning the request for clarification on this point, highlighted by the reviewer, we specify that in some cases the plants have been included in Table 2 because they are known and cited by informants with their dialectal names. In addition, the dialectal names of the local flora are the basis to calculate the EPI (Ethnophytonomic index) index, that allows to estimate the wealth of the people's knowledge about the local plant species, as reported in both Results and Discussion. Anyway, for a better understanding, in all these cases, reported below, we have added in Table 2 the following phrase: “Plant whose dialectal name is known”.

Lonicera caerulea L. subsp. caerulea: if there is no ethnobotanical uses why it is mentioned? See EPI

Agrostemma githago L.: if there is no ethnobotanical uses why it is mentioned? See EPI

Colchicum bulbocodium Ker Gawl.: if there is no ethnobotanical uses why it is mentioned? See EPI

Convolvulus arvensis L.: if there is no ethnobotanical uses why it is mentioned? See EPI

Hippophae rhamnoides L.: omit 'syn Elaeagnus rhamnoides (L.) A. Nelson' (this synonym has not been used for a long time) Done: The synonymous has been omitted

Juncus jacquinii L.: if there is no ethnobotanical uses why it is mentioned? See EPI

Salvia rosmarinus Spenn.: please, put 'Rosmarinus officinalis L.' as the main name and 'Salvia rosmarinus Spenn.' as a synonym We would like it too, but the nomenclature follows Plants of the World 

Rhinanthus alectorolophus (Scop.) Pollich: if there is no ethnobotanical uses why it is mentioned? See EPI

Abies alba Mill.: if there is no ethnobotanical uses why it is mentioned? See EPI

Primula pedemontana E.Thomas ex Gaudin: if there is no ethnobotanical uses why it is mentioned? See EPI

Pulsatilla alpina (L.) Delarbre: if there is no ethnobotanical uses why it is mentioned? See EPI

Trollius europaeus L.: if there is no ethnobotanical uses why it is mentioned? See EPI

Alchemilla vulgaris L.: please, put 'Alchemilla xanthochlora Rothm.' as the main name and 'A. vulgaris L.' as a synonym Done

Aria edulis (Willd.) M.Roem.: please, omit Aria edulis (Willd.) M.Roem. and keep only Sorbus aria (L.) Crantz. (synonym Aria edulis (Willd.) M.Roem. has not been used for a long time) We would like it too, but the nomenclature follows Plants of the World 

Rosa x alba L.: change letter x with symbol × Done

Rosa x centifolia L.: change letter x with symbol × Done

Sambucus racemosa L.: if there is no ethnobotanical uses why it is mentioned? See EPI

Line 117. Add space (row) after text below Figure 1. Done

Line 133. Add space (row) before Table 3.  Done

Table 3 Change 'Imperatoria ostruthium' with valid name 'Peucedanum ostruthium' Done

Delete point after 'genipi.' Done

Line 138: change 'Imperatoria ostruthium' with valid name 'Peucedanum ostruthium' Done 

Lines 154–154: change 'Alchemilla vulgaris' with valid name 'Alchemilla xanthochlora' Done 

Line 213: change 'J. communis' into 'Juniperus communis' (to be uniform with other plant names in this paragraph) Done 

Line 222: it shoud be: Rosa sp., (add point) Done 

Line 238. Add space (row) after text below Figure 3.  Done

Line 246. Change 'L.' into 'Larix' Done 

Line 246. Change 'P.' into 'Picea' Done 

Line 246. Change 'Pcembra' into 'Pinus cembra' Done 

Line 249: change 'vulgaris.' into 'vulgaris,' Done 

Line 266: change 'T. officinale' into 'Taraxacum officinale' Done 

Line 279: change 'Ribes petraeum' into 'R. petraeum' Done 

Line 288: change 'cereal' into 'cereale' Done 

Line 299: change 'U. dioica' into 'Urtica dioica' Done

Line 299: change 'A. genipi' into 'Artemisia genipi' Done

Line 303: change 'B. rapa' into 'Brassica rapa' Done

Line 303: change 'S. tuberosum' into 'Solanum tuberosum' Done 

Line 304: it shoud be: Triticum sp., (add point) Done

Line 309: change 'V. calcarata' into 'Viola calcarata' Done

Line 311: change 'L. usitatissimum' into 'Linum usitatissimum' Done

Line 314: change 'Cardus' into 'Carduus' Done 

Line 317: change 'G. punctata' into 'Gentiana punctata' Done

Line 318: What is A. montanaArnica montana? Done.

Line 321: change 'T. officinaleT. pratense and U. dioica' into ''Taraxacum officinaleTrifolium pratense and Urtica dioica' Done 

Lines 331, 338, 340, 352, 358, 423, 434, 451, 455, 593: as above to avoid confusion among so many species Done

Line 387: Rosa x alba L.: change letter x with symbol × Done

Lines 438, 442, 594: change 'I. ostruthium' with valid name 'Peucedanum ostruthium' Done 

Reviewer 5 Report

Dear Authors,

I am impressed by huge efforts which was done to perform the study, screen the literature, interpret the data and finally to write the manuscript.

I didn't find any gross errors in the manuscript, my only comment is perhaps that it is too long ?

I have also one small question: page 13, line 51, You wrote:

"Ethnobotanical data were collected during summer over two consecutive years (2017 - 2019)..." but it looks like the survey was done through 3 years: 2017, 2018 and 2019 ? Please clarify...

Author Response

We thank the reviewer for his positive comments on our paper and his useful suggestion. We have corrected clarifying that the survey was done through three years. We recognize that the work is very long but considering the amount of information collected it has not been possible to summarize it further. Many thanks for acknowledging the extensive effort behind the data presented.

Round 2

Reviewer 2 Report

Given the methodology, it might have been better if this article had been a review article, but I agree with the other reviewers.
After reviewing the latest version of the article, revised as directed by the other reviewers, I propose to accept the article.